# Multiscale networks in multiple sclerosis

**Keith E. Kennedy**[1], Nicole Kerlero de Rosbo[2,3], Antonio Uccelli[2], Maria Cellerino[2], Federico Ivaldi[2], Paola Contini[2], Raffaele De Palma[2], Hanne F. Harbo[4,5], Tone Berge[5,6], Steffan D. Bos[4,5], Einar A. Høgestøl[4,5], Synne Brune-Ingebretsen[4,5], Sigrid A. de Rodez Benavent[4,5], Friedemann Paul[7], Alexander U. Brandt[7,8], Priscilla Bäcker-Koduah[7], Janina Behrens[7], Joseph Kuchling[7], Susanna Asseyer[7], Michael Scheel[7], Claudia Chien[7], Hanna Zimmermann[7], Seyedamirhosein Motamedi[7], Josef Kauer-Bonin[7], Julio Saez-Rodriguez[9], Melanie Rinas[9], Leonidas G. Alexopoulos[10,11], Magi Andorra[12], Sara Llufriu[12], Albert Saiz[12], Yolanda Blanco[12], Eloy Martinez-Heras[12], Elisabeth Solana[12], Irene Pulido-Valdeolivas[12], Elena H. Martinez-Lapiscina[12], Jordi Garcia-Ojalvo[1‡*], Pablo Villoslada[1,13‡*]

**1** Department of Medicine and Life Sciences, Universitat Pompeu Fabra, Barcelona, Spain, **2** Department of Neurology, Ospedale Policlinico San Martino-IRCCS and Department of Neurosciences, Rehabilitation, Ophthalmology, Genetics, Maternal and Child Health, University of Genoa, Genoa Italy, **3** TomaLab, Institute of Nanotechnology, Consiglio Nazionale delle Ricerche (CNR), Rome, Italy, **4** Department of Neurology, University of Oslo, Oslo, Norway, **5** Department of Neurology, Oslo University Hospital, Oslo, Norway, **6** Oslo Metropolitan University, Oslo, Norway, **7** Department of Neurology, Charité—Universitätsmedizin Berlin, Corporate Member of Freie Universität Berlin and Humboldt-Universität zu Berlin, and Max Delbrueck Center for Molecular Medicine, Berlin, Germany, **8** Department of Neurology, University of California, Irvine, California, United States of America, **9** Institute for Computational Biomedicine, University of Heidelberg, Heidelberg, Germany, **10** ProtATonce Ltd, Athens, Greece, **11** School of Mechanical Engineering, National Technical University of Athens, Zografou, Greece, **12** Center of Neuroimmunology, Hospital Clinic Barcelona, and Institut d'Investigacions Biomediques August Pi i Sunyer, Barcelona, Spain, **13** Department of Neurology, Hospital del Mar Research Institute, Barcelona, Spain

‡ These authors are joint senior authors on this work.
* jordi.g.ojalvo@upf.edu (JGO); pablo.villoslada@upf.edu (PV)

## Abstract

Complex diseases such as Multiple Sclerosis (MS) cover a wide range of biological scales, from genes and proteins to cells and tissues, up to the full organism. In fact, any phenotype for an organism is dictated by the interplay among these scales. We conducted a multilayer network analysis and deep phenotyping with multi-omics data (genomics, phosphoproteomics and cytomics), brain and retinal imaging, and clinical data, obtained from a multicenter prospective cohort of 328 patients and 90 healthy controls. Multilayer networks were constructed using mutual information for topological analysis, and Boolean simulations were constructed using Pearson correlation to identified paths within and among all layers. The path more commonly found from the Boolean simulations connects protein MK03, with total T cells, the thickness of the retinal nerve fiber layer (RNFL), and the walking speed. This path contains nodes involved in protein phosphorylation, glial cell differentiation, and regulation of stress-activated MAPK cascade, among others. Specific paths identified were subsequently analyzed by flow cytometry at the single-cell level. Combinations of several proteins (GSK3AB, HSBP1 or RS6) and immune cells (Th17, Th1 non-classic, CD8, CD8 Treg, CD56 neg, and B memory) were part of the paths explaining the clinical phenotype. The advantage of the path identified from the Boolean simulations is that it connects information about these known biological pathways with the layers at higher scales (retina damage and

keithtopher.github.io/single_networks/#/ and
https://keithtopher.github.io/combo_networks/#/)
and paths (https://keithtopher.github.io/fivelayer_
pathways/.). Sequence data has been deposited at
the European Genome-phenome Archive (EGA),
under accession number EGAS00001007145
(https://ega-archive.org/studies/
EGAS00001007145).

**Funding:** This work was supported by the Spanish
Ministry of Science and Innovation and FEDER,
under project PID2021-127311NB-I00, the
European Commission (Horizon 2020 Framework
Programme, ERACOSYSMED ERA-Net program,
Sys4MS project, id:43 to PV); Instituto de Salud
Carlos III, Spain (AC1500052 to PV); the Italian
Ministry of Health (WFR-PER-2013-02361136 to
AU); the German Ministry of Science (Deutsches
Teilprojekt B "Förderkennzeichen: 031L0083B to
FP) and the Norwegian Research Council (project
257955 to HHH). J.G-O. was also supported by the
Maria de Maeztu Programme for Units of
Excellence in R&D (grant CEX2018-000792-M),
and by the Generalitat de Catalunya (ICREA
Academia programme). The funders had no role in
study design, data collection and analysis, decision
to publish, or preparation of the manuscript.

**Competing interests:** We have read the journal's
policy and the authors of this manuscript have the
following competing interests: KK reports no
disclosures. NKdR reports no disclosures. AU
received grants and contracts from FISM, Novartis,
Biogen, Merck, Fondazione Cariplo, Italian Ministry
of Health, received honoraria, or consultation fees
from Biogen, Roche, Teva, Merck, Genzyme,
Novartis. FI reports no disclosures. MC reports no
disclosures. HFH has received honoraria for
lecturing or advice from Biogen, Merck, Roche,
Novartis and Sanofi. TB has received unrestricted
research grants from Biogen and Sanofi-Genzyme.
SDB reports no disclosures. EH received honoraria
for lecturing and advisory board activity from
Biogen, Merck and Sanofi-Genzyme and
unrestricted research grant from Merck. SBI
reports no disclosures. SAdRB reports no
disclosures. FP received honoraria and research
support from Alexion, Bayer, Biogen, Chugai,
Merck Serono, Novartis, Genzyme, MedImmune,
Shire, Teva, and serves on scientific advisory
boards for Alexion, MedImmune, and Novartis. He
has received funding from Deutsche
Forschungsgemeinschaft (DFG Exc 257),
Bundesministerium fu?r Bildung und Forschung
(Competence Network Multiple Sclerosis), Guthy
Jackson Charitable Foundation, EU Framework
Program 7, National Multiple Sclerosis Society of
the USA. AUB is named as inventor on multiple

disability). Overall, the identified paths provide a means to connect the molecular aspects of MS with the overall phenotype.

## Author summary

Complex diseases such as Multiple Sclerosis (MS) involve the contribution of a wide range of biological processes. We conducted a systems biology study of MS based on network analysis and deep phenotyping in a prospective cohort of patients with clinical, imaging, genetics, and omics assessments. The genes, proteins and cell paths explained variation in central nervous system damage, and in metrics of disease severity. Such multilayer paths explain the different phenotypes of the disease and can be developed as biomarkers of MS.

### Highlights

- Complex diseases involve the contribution of a wide range of biological processes at different scales. Multilayer network analysis allowed the study of the flow of information across scales.

- The validated path with the highest joint cross-correlation connects the protein MK03, previously associated with MS, with Total T cells, the thickness of the retinal nerve fiber layer (RNFL) and the timed 25 foot walk test (T25WT).

- Combinations of several proteins (GSK3AB, HSBP1 or RS6) and immune cells (Th17, Th1 non-classic, CD8, CD8 Treg, CD56 neg, and B memory) were part of the paths explaining the clinical phenotype.

## Introduction

Complex diseases involve the interaction of multiple biological scales, including tissues, cells, and molecules (genes, proteins, and metabolites), all of which regulate biological function and modulate the susceptibility to a given clinical phenotype. Although significant efforts have been devoted to understanding each of these levels, few attempts have succeeded in integrating multiple scales and the flow of information across them. Such integration would definitely improve our understanding of disease pathogenesis [1,2] and wellness [3]. Multilayer networks provide a framework to integrate complex biological data across different scales, which should allow us to understand the flow of biological information in health and disease [4–6]. This is especially important in diseases with a complex genetic and molecular basis, such as Multiple Sclerosis (MS).

MS is an autoimmune disease characterized by inflammatory attacks to the central nervous system (CNS), which damages the neural tissue and leads to significant disability [7]. The inflammation occurs in acute attacks as well as by chronic inflammation, defining the different clinical subtypes of the disease, namely relapsing-remitting (RRMS) and progressive (PMS). MS is an example of a complex disease, with different biological scales participating in its pathogenesis, including genetic factors [8], cellular signaling [9,10], adaptive and innate immunity [11,12], and CNS damage [13]. Additionally, the interplay between these various components is modulated by environmental factors [14,15], with viral infections and especially the Epstein-

patents and patents pending owned by Charité - Universitätsmedizin Berlin and/or University of California Irvine for visual computing-based motor function analysis, multiple sclerosis serum biomarkers, and retinal image analysis. He is cofounder and holds shares of Motognosis GmbH and Nocturne GmbH. He serves on the executive board and is Treasurer/Secretary of IMSVISUAL. He received research support from BMWi, BMBF, NIH ICTS, the Kathleen C. Moore Foundation and the Guthy- Jackson Charitable Foundation. Priscilla Ba?cker-Koduah is funded by the DFG Excellence grant to FP (DFG exc 257) and is a Junior scholar of the Einstein Foundation. CC received honoraria for speaking from Bayer and research funding from Novartis, unrelated to this study. SA received a conference grant from Celgene and honoraria for speaking from Alexion, Bayer and Roche. JB reports no disclosures. JSR declares funding from GSK & Sanofi and fees from Travere Therapeutics & Singularity Bio. MR reports no disclosures. LGA is founder and hold stocks at ProtATonce. MA is an employee of Hoffman-La Roche AG, yet this article is related to his activity at the Hospital Clinic of Barcelona. EHML is an employee of the European Medicines Agency (Human Medicines) since 16 April 2019, yet this article is related to her activity at the Hospital Clinic of Barcelona and consequently, it does not in any way represent the views of the Agency or its Committees. SL received compensation for consulting services and speaker honoraria from Biogen Idec, Novartis, TEVA, Genzyme, Sanofi and Merck. AS received compensation for consulting services and speaker honoraria from Bayer-Schering, Merck- Serono, Biogen-Idec, Sanofi-Aventis, TEVA, Novartis and Roche. EMH reports no disclosures. Elisabeth Solana received travel reimbursement from Sanofi and ECTRIMS and reports personal fees from Roche Spain. IPV is currently an employee of UCB pharma, yet this article is related to her activity at the Hospital Clinic of Barcelona. She has received travel reimbursement from Roche Spain and Genzyme-Sanofi, European Academy of Neurology, and European Committee for Treatment and Research in Multiple Sclerosis for international and national meetings over the last 3 years; she holds a patent for an affordable eye-tracking system to measure eye movement in neurologic diseases, and she holds stock in Aura Innovative Robotics. JGO reports no disclosures. PV has received consultancy fees and held stocks from Accure Therapeutics SL, Attune Neurosciences Inc, Spiral Therapeutics Inc, QMenta Inc, CLight Inc, NeuroPrex Inc, StimuSIL and Adhera Health Inc

Barr virus being the main triggers [16]. As a result, the MS phenotype of neurological disability is very heterogeneous and difficult to predict [7,17], creating significant limitations for patient care. As an example of the difficulty of finding biological determinants of MS, although more than 200 genetic polymorphisms have been associated with MS susceptibility, their contribution to the clinical phenotype is small and remains to be clarified [18]. Similarly, many studies have attempted to identify biomarkers of the clinical course and prognosis of the disease, including oligoclonal bands, neurofilament light chain protein, brain or spinal cord volume or retinal thickness, but few have been validated, and even their individual predictive ability is small, making their use in clinical practice limited [19].

Several studies have attempted to integrate biological networks in MS, mainly at the genetic level [20–23]. Those studies addressed the biomolecular aspects of the disease (genes and proteins), but they did not describe the relation of those features with tissue damage or clinical disability. In contrast, our approach focuses on bridging the gap between the microscopic and macroscopic scales of MS to better explain the endotype-phenotype relationship. To that end, we use multilayer network analysis to assess how information flows across biological scales, and to identify multiscale paths that contribute to explain the phenotype of MS.

Within the umbrella of the Sys4MS project [24], we recruited a multicenter prospective cohort of 328 patients with MS and 90 healthy subjects with a two-year follow-up and performed deep phenotyping by collecting multi-omics data, imaging, and clinical outcomes from the subjects. This collection provided data on five biological layers: (1) genes (risk alleles), (2) phosphoproteins (mostly kinases), (3) immune cells, (4) CNS tissue (imaging), and finally (5) the clinical phenotype (**Fig 1A**). We conducted two network analyses, one for the structural or topological analysis (using mutual information) and another based on the dynamics of the network (using Boolean modeling and Pearson correlation, that provides positive and negative values for the edges) (**S1 Fig**). Network generation was first applied to each of these layers individually, using mutual information to capture linear and non-linear dependencies between the elements of each layer (**Fig 1B–1F**) before the layers were interconnected (**Fig 1G**). Considering the non-linear relationships in this way allows us to incorporate diverse data-types across the layers. Our approach is hypothesis-based, rather than data-based: First, we make use of a set of single nucleotide polymorphisms (SNPs) derived from DNA arrays associated with MS, proteins and immune cell subtypes already known to be associated with MS [9,24–26]. Second, we consider the transfer of information from genes to proteins and cell layers, which will define the CNS tissue (imaging) and clinical outcomes as the phenotype (**Fig 1G**). In order to obtain functional information from the network models, dynamical simulations using Boolean network modeling were used to identify several paths spanning these five layers.

## Results

The focus of the results is on the paths between the genes, proteins, cells and the phenotype (imaging and clinical variables). Each step below shows how the paths were identified and which sources tend to be more strongly connected with the phenotype. First, descriptive information about the data is given, then the networks of the layers are constructed, then Boolean simulations are run, and finally the top paths are selected and validated.

### Deep phenotyping: multi-omics, imaging, and clinical data from MS patients

We recruited 328 MS patients (age 41±10 years, 70% female) at four centers throughout Europe, corresponding to the Sys4MS cohort (**Table 1**). Of these, 271 patients (82%) had RRMS, and 57 (17%) had PMS. We also recruited 90 healthy controls (HCs) matched by sex

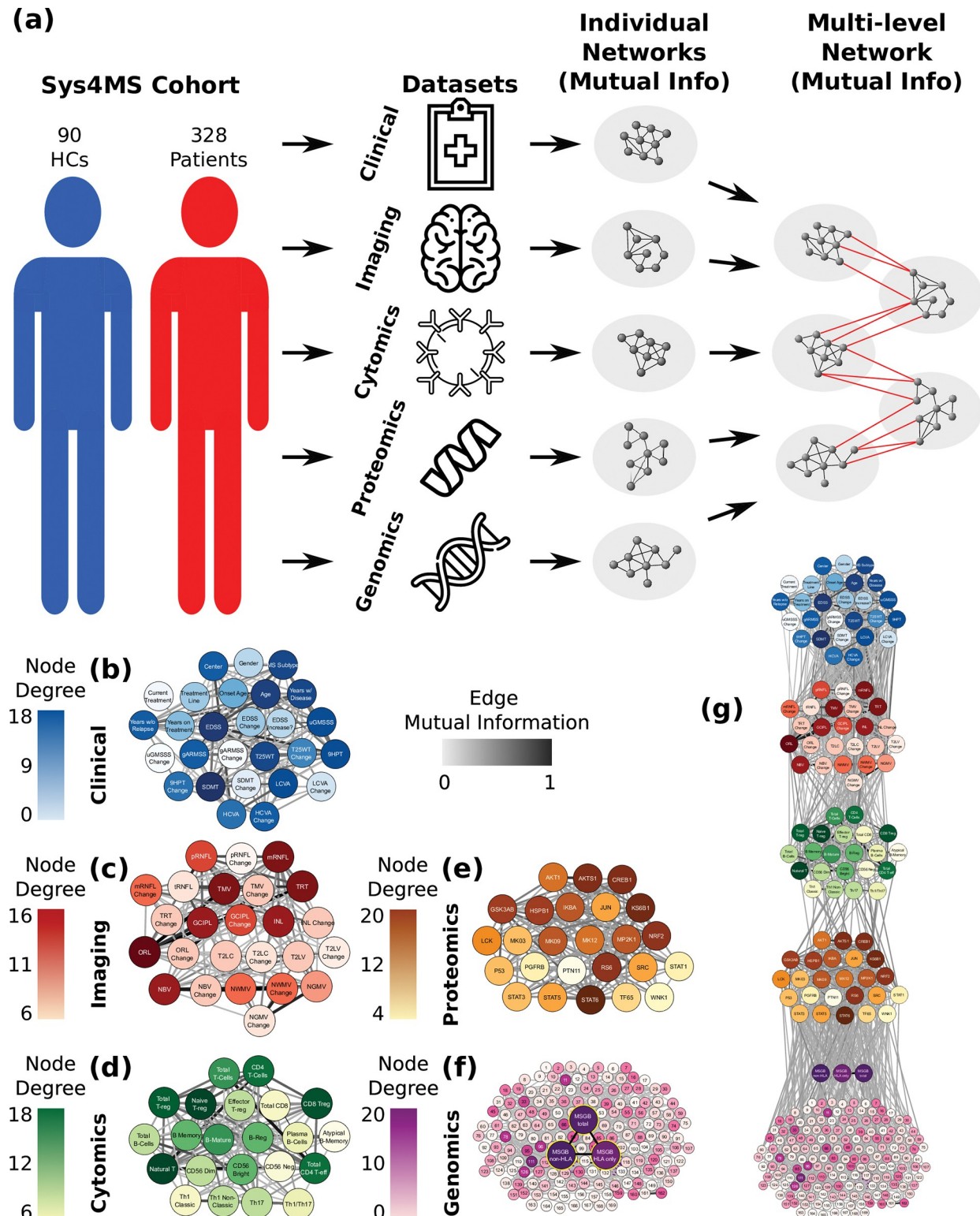

**Fig 1. Building multilayer networks using multi-omics, imaging, and clinical data.** (a) Illustration of network construction. The data from each layer is taken from the cohorts and used to create networks, where the nodes are the elements in the dataset (genomics (SNPs), phosphoproteomics, cytomics, CNS tissue imaging, and clinical data), and the edges correspond to the mutual information between element pairs across all subjects. Once individual networks are created, they are linked together, again using mutual information, following a hierarchy that connects each layer successively, starting with genomics and working up to the phenotypic (clinical) layer. (b-f) Topology of individual layer

networks from the experimental data. In each of the networks, the degree of each node is color-coded, with higher degrees in darker colors. The edge weights are coded in gray scale in a similar manner, with a darker edge representing a higher weight, and thus a higher correlation between nodes. The genomics network was enriched with the previous knowledge on regulatory networks (f) and included the MS genetic burden scores (g). In the combined five-layer network, the layers are connected using the hierarchy described above, with genomics at the bottom and clinical phenotype at the top. These networks are meant to show the nodes that are more highly correlated with other nodes in the network. They provide the base to examine the topological structure of the overall multilayer network. High resolution network representations for single-layer networks are available at Github link https://keithtopher.github.io/single_networks/#/ and for multilayer networks at https://keithtopher.github.io/combo_networks/#/. Icons used in the figure are open source from onlinewebfonts.com, flaticon.com and icons8.com.

and age with the RRMS group. The patients had a mean disease duration of 10 (SD 8) years, and median Expanded Disability Status Scale (EDSS) of 2.0 (range: 0–8). Regarding the use of disease modifying drugs (DMD) at baseline, 70% of patients were treated, 44% with low-efficacy therapies, and 26% with high-efficacy therapies (see **Methods** for drug definition). By the second year of follow-up (mean follow-up 1.98 ± 0.94 years, n = 274), two RRMS cases progressed to PMS, 22 patients started new therapies (cladribine: 1; fingolimod: 2; glatiramer acetate: 4; ocrelizumab: 9; rituximab: 2; teriflunomide: 4) and 17 changed from low to high-efficacy therapies. Imaging data consisted of both brain magnetic resonance imaging (MRI) and retina optical coherence tomography (OCT) (**Table 1**).

We conducted a genetic analysis in both MS cases and controls. From the 700,000 SNPs assessed in the DNA array, we imputed 152 SNPs associated with MS [8], along with 17 additional SNPs corresponding to HLA-class II alleles. We calculated the polygenic risk score, namely the MS genetic burden score [27] (MSGB) for all 169 SNPs, together with partial MSGB scores for only the 17 HLA SNPs (MSGB$^{HLA}$), and for the 152 MS associated SNPs excluding the HLA alleles (MSGB$^{non-HLA}$). As expected, the total MSGB score was significantly higher (p = $3.4 \times 10^{-8}$) in patients (4.23) than in HCs (3.2). Similar results were observed in the partial scores, with MSGB$^{HLA}$ of 1.57 in patients and 0.95 in HCs (p = $1.6 \times 10^{-4}$) and MSGB$^{non-HLA}$ of 2.6 in patients and 2.2 in HCs (p = $6.8 \times 10^{-5}$).

Flow cytometry analysis was carried out at baseline in peripheral blood mononuclear cells (PBMCs) from the first 227 patients and 82 HC. Results from the cytometry analysis in this cohort are described in detail elsewhere (24). Briefly, untreated RRMS patients showed significantly higher frequencies of Th17 cells and lower frequencies of B-memory/B-regulatory cells, as well as higher percentages of mature B cells in patients with PMS compared with HCs. Fingolimod treatment induced a decrease in total CD4+ T cells and mature and memory B cells and increases in CD4+, CD8+ T-regulatory and B-regulatory cells [24]. Finally, the phospho-proteomic analysis was carried out by conducting ex-vivo assays in PBMCs and quantified using xMAP assays on the first 148 patients at baseline as described before [26, 28], showing higher levels of phosphorylated IKBA, JUN, KSGB1, MK03, RS6, STAT3 and STAT6 in MS patients compared to controls (**S1 File**).

## Multilayer networks in MS

All data used in this study is coming from the study's cohort except for the reference Genomic Regulatory Network used to embed the SNP data. We built networks for each of the five layers (genetics, phosphoproteomics, cytomics, tissue/imaging and clinical variables) using mutual information to define connections between pairs of elements within each layer (**Fig 1**, see **Methods**). At this stage, mutual information is used due to its ability to detect linear and nonlinear correlations between two elements. For example, in the proteomics layer two proteins are connected to each other with a weight equal to the normalized mutual information between their phosphorylation levels. A threshold was used to determine whether the correlation for a given pair was high enough to define an edge. The threshold works by comparing

**Table 1. Sys4MS cohort: Clinical and imaging variables of MS patients and healthy controls.** Disability scales are shown as the mean ± SD, except for the EDSS which is displayed as the median (range).

| | MS baseline n = 328 | MS 2-year follow-up n = 278 | HC n = 90 |
|---|---|---|---|
| Age | 41±10 | 45±9.81 | 36.98 ± 11.47 |
| Female, n (%) | 229 (70%) | 194 (70%) | 63 (70%) |
| Age at disease onset (years) | 31 ± 9 | 31±9 | – |
| Disease duration (years) | 10 ± 8 | 12.9±8.16 | – |
| Relapse-Remitting MS (RRMS) Primary Progressive MS (PPMS) Secondary Progressive MS (SPMS) | 271 29 28 | 228 25 25 | – |
| Expanded Disability Status Scale (EDSS) | 2.0 (0–8.0) | 2.0 (0–8.0) | – |
| MS Severity Scale (MSSS) | 3.6 ± 2.2 | 3.25±2.35 | – |
| Age-Related MS Severity Scale (ARMSS) | 3.9 ± 2.1 | 3.56±2.26 | – |
| Timed 25-Feet Walking Test (T25WT) (sec) | 6.93 ± 6.6 | 5.67±4.97 | – |
| 9-Hole Peg Test (9HPT) (sec) | 21.2 ± 6.5 | 21.9±5.92 | – |
| Symbol Digit Modality Test (SDMT) (# symbols) | 53.8 ± 13.5 | 53.5±13.3 | – |
| 2.5% Sloan Letter Acuity (SL25) (# letters) | 29.1 ± 13.4 | 26.7±13.5 | – |
| High-Contrast Visual Acuity (HCVA) (LogMAR) | 0.03 ± 0.36 | -0.11±0.44 | – |
| Disease Modifying Drug (DMD) Untreated | 91 | 72 | – |
| Interferon beta | 43 | 19 | – |
| Glatiramer acetate | 39 | 24 | – |
| Teriflunomide | 28 | 21 | – |
| Fingolimod | 38 | 33 | – |
| Dimethyl-Fumarate | 35 | 37 | – |
| Natalizumab | 29 | 24 | – |
| Other high-efficacy DMD* | 19 | 43 | – |
| **Brain MRI** | | | **baseline** |
| # Gadolinium lesions | 0.1 ± 0.5 | NA** | NA |
| T2 lesion volume (T2LV) ($cm^3$) | 8.17 ± 10.5 | 9.32±11 | NA |
| Normalized Brain Volume (NBV) ($cm^3$) | 1,509 ± 91 | 1,454±70.2 | 1,473±109 |
| Normalized Gray Matter Volume (NGMV) ($cm^3$) | 792 ± 65 | 779 ±49.5 | 751±63.7 |
| Normalized White Matter Volume (NWMV) ($cm^3$) | 716 ± 68 | 676 ±43.5 | 721±111 |
| **OCT (mean of both eyes w/o previous optic neuritis)** | | | **baseline** |
| Peripapillary Retinal Nerve Fiber Layer (pRNFL) (μm) | 100 ± 12.7 | 101±12.1 | 100±9.6 |
| Macular Retinal Nerve Fiber Layer (mRNFL) (μm) | 39.6 ± 4.9 | 39.6±4.31 | 41.9±6.5 |
| Ganglion Cell Plus Inner Plexiform Layer (GCIPL) (μm) | 65.6 ± 8.3 | 65.7±7.08 | 68.5±6 |
| Inner Nuclear Layer (INL) (μm) | 31.5 ± 2.8 | 31.5±2.77 | 41.1±8.8 |
| Outer Nuclear Layer (ORL) (μm) | 146.1 ± 9.5 | 147±8.39 | 149±19.9 |

*Other DMD baseline: alemtuzumab: 9, rituximab: 7, ocrelizumab: 1, daclizumab: 2; year 2: alemtuzumab: 13, rituximab: 11, ocrelizumab: 16, cladribine: 3

**MRI studies for the follow-up did not include gadolinium administration.

the real mutual information value of a pair of nodes to a surrogate distribution of mutual information values calculated from random permutations of the data.

The genetic network was considered in two ways: first, at the level of the individual SNPs separately and second, utilizing previous information from the Gene Regulatory Network Database [29] and mapped to the MS associated SNPs (see **Methods**); and second, grouped together in the three MSGB scores defined above. The proteomic network includes 25 kinases,

and the cytomics network 22 immune subpopulations (see **Methods** for the lists of proteins and cell subtypes). The imaging network included the main metrics of lesion load and brain volumes quantified by MRI, and the thickness of the retina layers analyzed by OCT. Finally, the clinical network contains demographic and clinical variables (number of relapses, disability scales and use of DMD) at baseline and after two-year follow-up, which give longitudinal changes in clinical outcomes (see **Methods** for a list of variables).

The networks seen in **Fig 1** shows which nodes are more highly connected with the other nodes in the datasets. Nodes with more connections (hubs) imply they are central to the information flow within each layer. Changes in one hub node can signal changes in many other nodes and serve as a means of prediction. For each layer, the hub nodes are shown in **Table 2**.

After the networks for each layer were built, we sought to connect the elements from the various layers together. Edges were again defined using mutual information, but this time two elements from any of the five layers were connected. At this step, hierarchy among the layers is ignored to gain a sense of how the network is connected regardless of scale. We analyzed the connectivity (density) between layers, this time between features of different layers. A statistical comparison between the connections within and between layers shows a non-negligible degree of network modularity, confirming the underlying multi-layer structure (**Fig 2**). The features within a layer are, on average, more strongly connected than those between layers. With the exception of genomics, the densities within a single layer were higher than those between layers, supporting the modularity of the multilayer network. Since the density analysis confirmed that nodes are preferentially attached within their layer, this suggests that mutual information is able to adequately manage the diversity of the data types. In summary, the topological analysis of the multilayer networks (using mutual information) shows that nodes are highly connected within the same layer than between layers, which indicates that the nature of the data determines the connectivity. Second, some layers showed higher connectivity than others (e.g., imaging compared to genomics), suggesting higher complexity of the interactions while going up the hierarchy. The topological multilayer network for the different disease subtypes (RRMS vs PMS), disease severity (mild vs severe) and disease modifying therapies is shown in **S2 Fig**.

## Dynamic network analysis identifies gene-protein-cell paths associated with phenotype

We next sought to integrate all the layers in paths that reflect the network dynamic interactions, in order to obtain a functional view of the information flow across layers. To that end,

**Table 2. Hubs within each layer network.**

| Layer | Hub Nodes |
|---|---|
| Clinical | age<br>expanded disability status score (EDSS)<br>symbol digit modality test (SDMT) |
| Imaging | outer nuclear layer (ORL)<br>total macular volume (TMV)<br>macular retinal nerve fiber layer (mRNFL) |
| Cytomics | Naive T-regulatory<br>Natural T Cells<br>CD8 T-regulatory |
| Proteomics | STAT6<br>KS6B1<br>CREB1 |
| Genomics | MSGB total<br>MSGB HLA only<br>MSGB non-HLA |

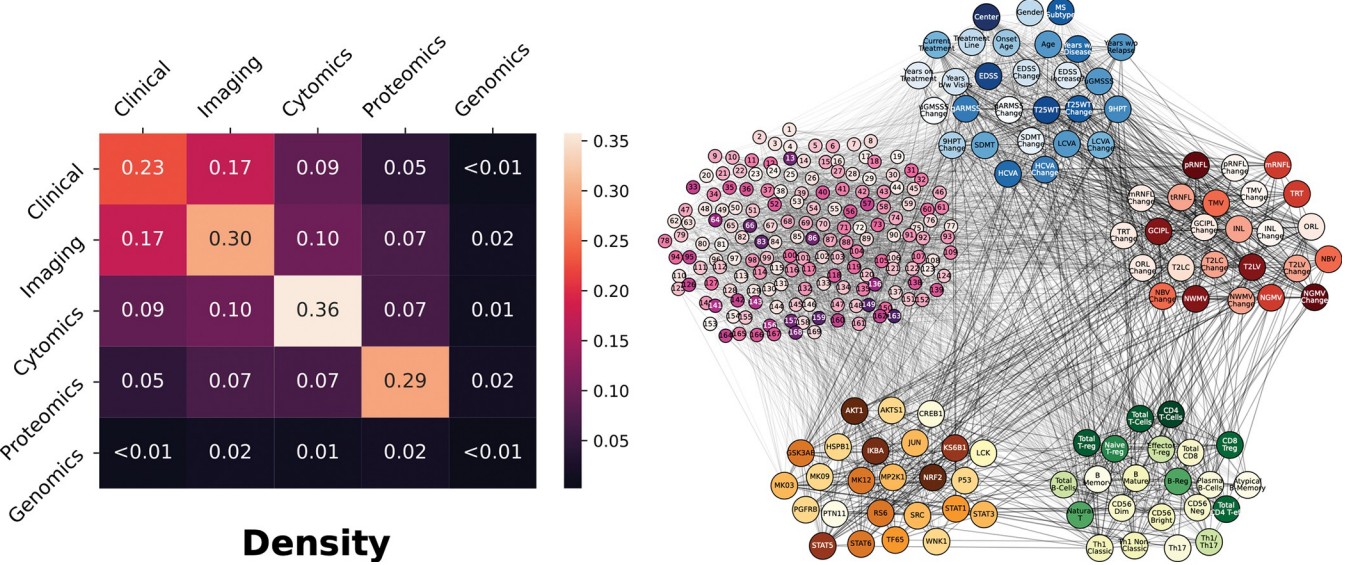

**Fig 2. Network densities within and between layers.** (*left*) The density for each layer was calculated as the ratio of the sum of the weights of all connections and the number of possible connections. The analysis was made using the 67 subjects with complete data in all 5 layers. (*right*) The network from which the density was calculated. Nodes from all layers were connected together, opposed to the network model with the hierarchy shown before. See high resolution network at https://keithtopher.github.io/combo_networks/#/.

we created a single network including all five layers with the same hierarchy described above. For the dynamic network analysis, instead of using mutual information, we used linear (Pearson) correlations, which allows us to distinguish between stimulatory or inhibitory edges (which is required for logic modeling) depending on whether the correlation r value is positive or negative, respectively (**Fig 3A**). We then conducted logic (Boolean) simulations to identify the causal logic backbone of the network [30,31]. Boolean simulations use knowledge of activating and inhibiting relationships between nodes; the exact chemical reactions between genes, proteins, cells and tissue are ignored, giving a qualitative description of the system [31]. The nodes of the network are considered to be in one of two states: active (e.g., high phosphorylation levels) or inactive (e.g., low phosphorylation levels). The states of all nodes are updated synchronously at each iteration of the simulation, either remaining in the same activation state as before, or flipping to the opposite state, depending on the activation states of its direct neighbors, and taking into consideration the weights of the corresponding connections (**Fig 3B**, see **Methods**).

We next wanted to study how perturbations in a given input such as the MSGB score (SNPs could not be used for Boolean simulations because the impossibility of changing between alleles), protein or cell type travel through the network and ultimately affect a given phenotype (output). To that end, we performed Boolean simulations in which the input node was periodically driven from an active to an inactive state and back, and the response of all nodes in the network (**Fig 3C**) was quantified by computing the temporal cross-correlation function between their time-varying state and the dynamic input signal [31]. We then identified those paths across the network that are formed by pairs of nodes with the highest temporal cross-correlation between their signals. These paths represent how information flows from a given input to the output (e.g., from MSGB-non-HLA to clinical scale EDSS in the example in **Fig 3E**). They do not necessarily represent physical interactions among nodes (e.g., protein-protein interactions), but rather groups of nodes that co-vary statistically with each other more strongly than the rest of the network.

**(a) Multi-Layer Network (Pearson)**

positive correlation

negative correlation

Layer 1　　　　　Layer 2

**(c) Simulations with Gene Input**

5% Noise

Clinical

Imaging

Cytomics

Proteomics

MSGB non-HLA

Genomics

Active

Inactive

0　　　50　　　100

Timesteps

**(b) Boolean Dynamics**

Time

*Boolean State*

○ - Active

● - Inactive

*Pearson Corr.*

▬ - Positive

▬ - Negative

**(d) Calculation of Pathway Cross Correlation**

1/C1　　1/C2　　1/C3

Path Score = 1/C1 + 1/C2 + 1/C3

**(e) Pathway Identification**

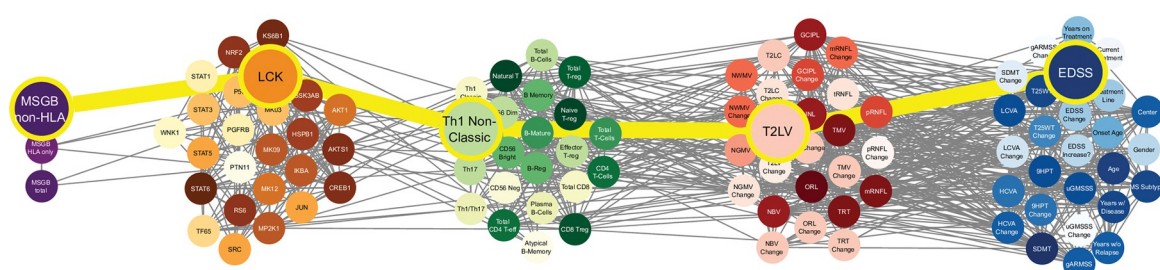

**Fig 3. Dynamic network analysis: identification of gene-protein-cell paths.** (a) Networks are constructed using all five layers. The nodes are the same as in the networks above (Fig 1), but now the edges are defined by the Pearson correlation, where the weights represent the Pearson coefficient, which can be either positive or negative. (b) Boolean dynamics are applied to the networks, where the activation state of the nodes changes based on the total sum of the edge weights of its direct neighbors (considering the signs of the connections). (c) Boolean simulations are run where the various nodes, in the example MSGB non-HLA, are used as the input signal, and

the simulation was run with 5% noise (see **Methods** for noise analysis). (d) The cross-correlation coefficient (Cn) is calculated between the signals for each pair of connected nodes. A path score is calculated for all possible paths, defined as the sum of the inverses of the cross-correlation coefficients between all pairs of consecutive nodes constituting a given path. (e) Finally, a path is identified by using a shortest path algorithm which is based on its path score (see **Methods**).

For each of the 3,350 combinations of inputs and outputs (3 MSGB scores, 25 proteins, and 22 cell types as inputs, and 22 cell types, 25 imaging variables, 20 clinical variables as outputs), we selected the top ten paths with highest joint cross-correlation values between their constituent nodes (see **Methods** and **S2 File**). **Figs 4–6** shows these paths for the three inputs (MSGB, phosphoproteomics and cytomics) and outputs (imaging and clinical) pairs for MS patients.

To assess the specificity of the Boolean simulations, the network was permuted to identify negative control paths. The edges were randomly swapped while preserving the original degree distribution of the network. The simulations were run with these permuted networks (100 total), and paths were identified. These paths were compared to those identified in the original

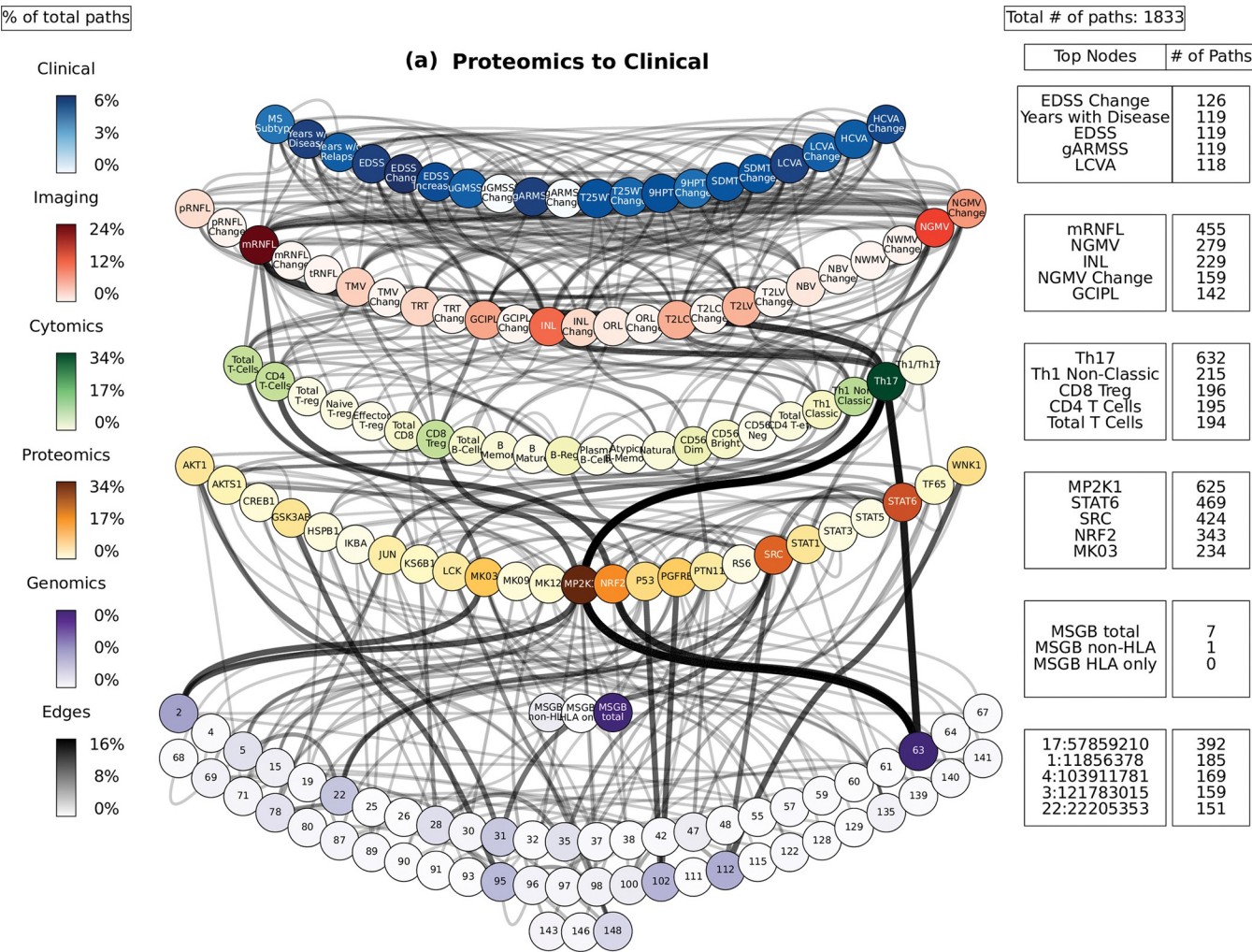

**Fig 4. Path analysis in MS patients.** Representations of the multi-layer paths identified from the Boolean simulations when the input started at the phosphoproteomics layer. The top paths (those that passed the test for negative controls) are shown for each input (gene, protein, or cell)-output (clinical phenotype) pair. The nodes for each layer are color-coded to represent the degree of a given node, i.e., the number of times the node appears in a path, as a percentage of the total number of paths. High resolution paths are available at https://keithtopher.github.io/fivelayer_pathways/.

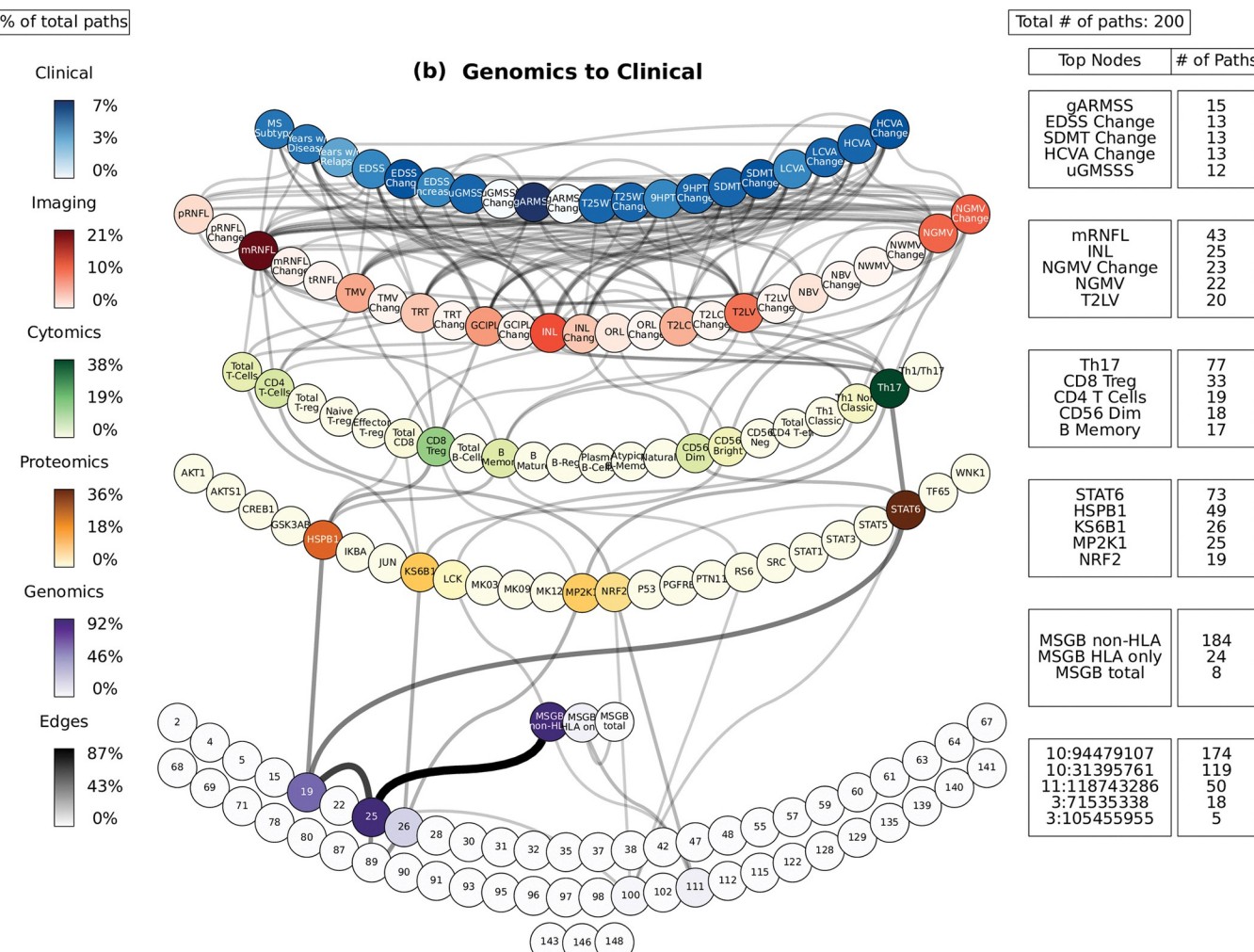

**Fig 5. Path analysis in MS patients.** Representations of the multi-layer paths identified from the Boolean simulations when the input started at the genomics layer. The top paths (those that passed the test for negative controls) are shown for each input (gene, protein, or cell)-output (clinical phenotype) pair. The nodes for each layer are color-coded to represent the degree of a given node, i.e., the number of times the node appears in a path, as a percentage of the total number of paths. High resolution paths are available at https://keithtopher.github.io/fivelayer_pathways/.

networks. We counted how many times a given path appeared in the permuted networks. Focus was placed on those pathways that were present in less than 1% of the permuted pathways. Out of 32,302 total paths identified from MS patients, there were 8,488 that appeared 0 times out of 100 in the permuted paths. The method for network permutation and path identification is illustrated in **Methods,** and results are shown as **S3 and S4 Files**.

Additionally, confidence intervals were calculated for each of the paths. The paths were identified from each of the 100 Boolean simulations individually (instead of using the mean of the cross-correlation values as before). These 100 simulations provide a distribution of path scores, giving the variance of the original path score. The path scores along with their confidence intervals are given in **S5 File**.

## Path analysis

For the path analysis we use the following notation: NODE 1 > NODE 2 > NODE 3, (in the case there are multiple nodes on the same layer along similar paths they appear as NODE

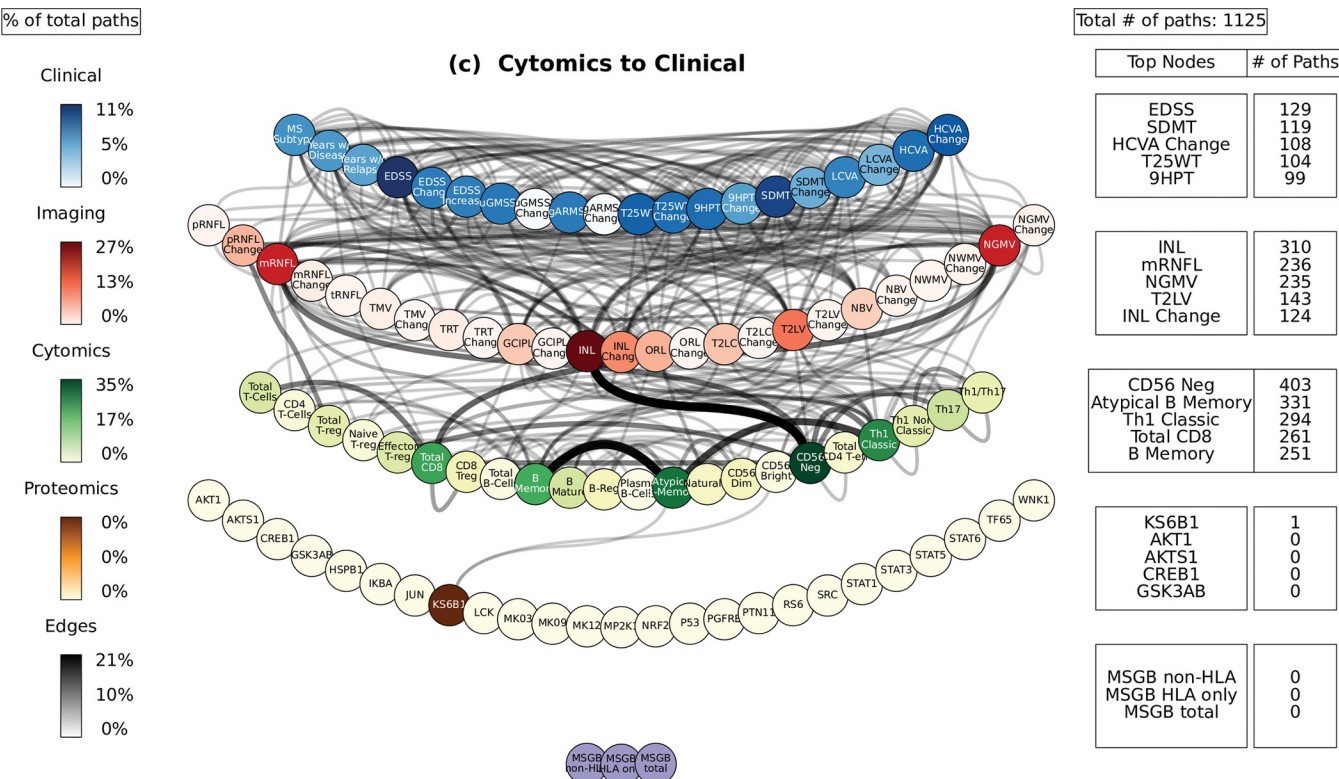

**Fig 6. Path analysis in MS patients.** Representations of the multi-layer paths identified from the Boolean simulations when the input started at the cytomics layer. The top paths (those that passed the test for negative controls) are shown for each input (gene, protein, or cell)-output (clinical phenotype) pair. The nodes for each layer are color-coded to represent the degree of a given node, i.e., the number of times the node appears in a path, as a percentage of the total number of paths. High resolution paths are available at https://keithtopher.github.io/fivelayer_pathways/.

1 > NODE 2—NODE 3 > NODE 4, where NODES 2 and 3 could be two proteins for example.) and the information flows from left to right, starting with the perturbation in the gene, protein, or cell respectively. MS cases show that the paths more commonly found from the Boolean simulations (darker color represents more connections or higher degree) were: (1) MP2K1 > Th17 > mRNFL > ARMSS (when the input is applied to the started in phosphoproteomics layer, **Fig 4**); (2) SNP25 (SNP10:94479107) > MSGB non-HLA > STAT6 > Th17 > mRNFL> ARMSS (when the input is applied to the genomics layer, **Fig 5**); (3) CD56 Neg > INL—mRNFL > EDSS—ARMSS (when the input is applied to the cytomics layer, **Fig 6**).

Perturbations in the protein layer (representing changes in the signaling cascades among cells) were linked with the severity of MS, this time with both the EDSS and ARMSS along with the HCVA, T25WT and the disease duration (**Fig 4**). **Figs 7–9A** shows a subset of the top paths identified with protein sources along with the Pearson correlation between nodes. The dynamic multilayer network for the different disease subtypes (RRMS vs PMS), disease severity (mild vs severe) and disease modifying therapies is shown in **S3 Fig**. More in depth details about validation of paths and their connections to biological pathways are given in **S9 File**.

Perturbations of the gene network (the MSGB, reflecting genetic variability contributing to the risk of developing MS) were linked with changes in the clinical outcomes (ARMSS, T25WT, 9HPT, HCVA, LCVA, and the EDSS) (**Fig 5**). **Fig 8** shows a subset of the top paths identified with protein sources along with the Pearson correlation between nodes. More in depth details about validation of paths and their connections to biological pathways are given

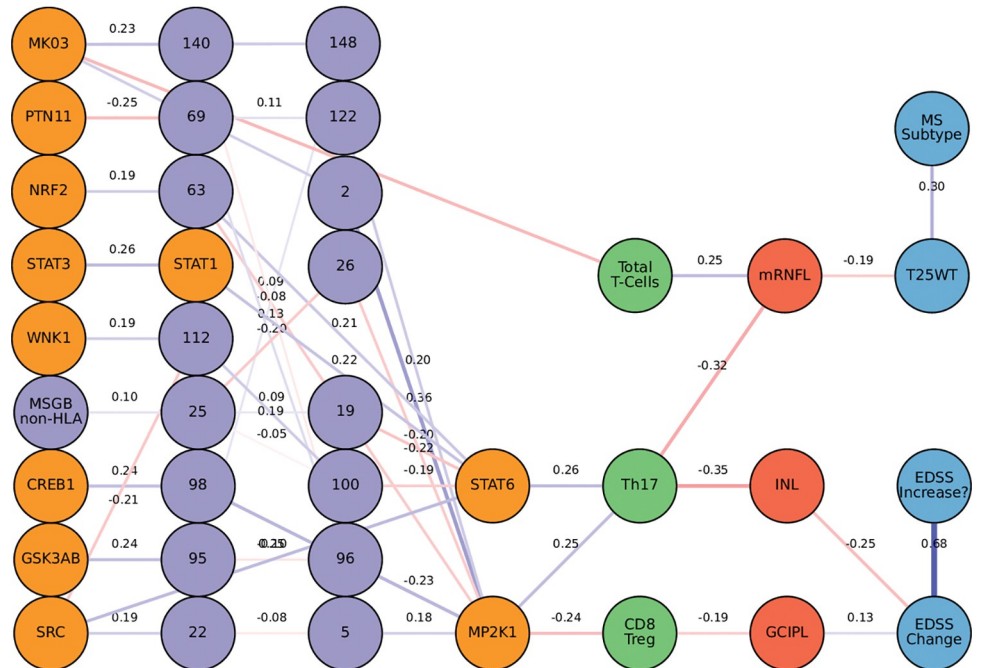

**Fig 7. The network of the top paths associated with MS.** paths starting at the proteomic layer.

in **S9 File**. Concerning the imaging layer, we found paths to the mRNFL (macular retinal nerve fiber layer) and NGMV (normalized gray matter volume). Perturbing the MSGB non-HLA was the source for the most paths at this level: 1) MSGB non-HLA > SNP10:94479107 > SNP11:118743286 > KS6B1—MP2K1; and 2) MSGB non-HLA > SNP10:94479107 > SNP10:31395761 > HSPB1—STAT6.

Perturbations at the cellular level (representing changes of immune cell subtypes frequency and activation) were connected again with changes in the EDSS as well as with the HCVA, SDMT, 9HPT, and T25WT (Fig 6). **Fig 9** shows a subset of the top paths identified with protein sources along with the Pearson correlation between nodes. More in depth details about validation of paths and their connections to biological pathways are given in **S6 File**.

## Paths predicting MS phenotype from single-cell data

In order to assess some of the paths identified in the study at the single-cell level, we conducted a cytometry analysis to assess levels of total and phosphorylated proteins in immune cell subtypes at the single-cell level and relate them to the clinical phenotype through linear regression models and path analysis. We analyzed the levels of the three phosphoproteins for which phospho-cytometry assays were available and that showed an adequate signal to noise ratio, namely GSK3AB, HSBP1 and RS6 (assays were not validated for the other proteins). We also analyzed the immune cell subpopulations most commonly present in such paths (CD4+, Treg, CD8+, B mature, B memory, Breg and Plasma cells). This approach allowed to assess experimentally the paths between an individual phosphoprotein in the selected immune cell subtype. The phosphorylation levels in immune subpopulations were assessed in a representative subgroup of 40

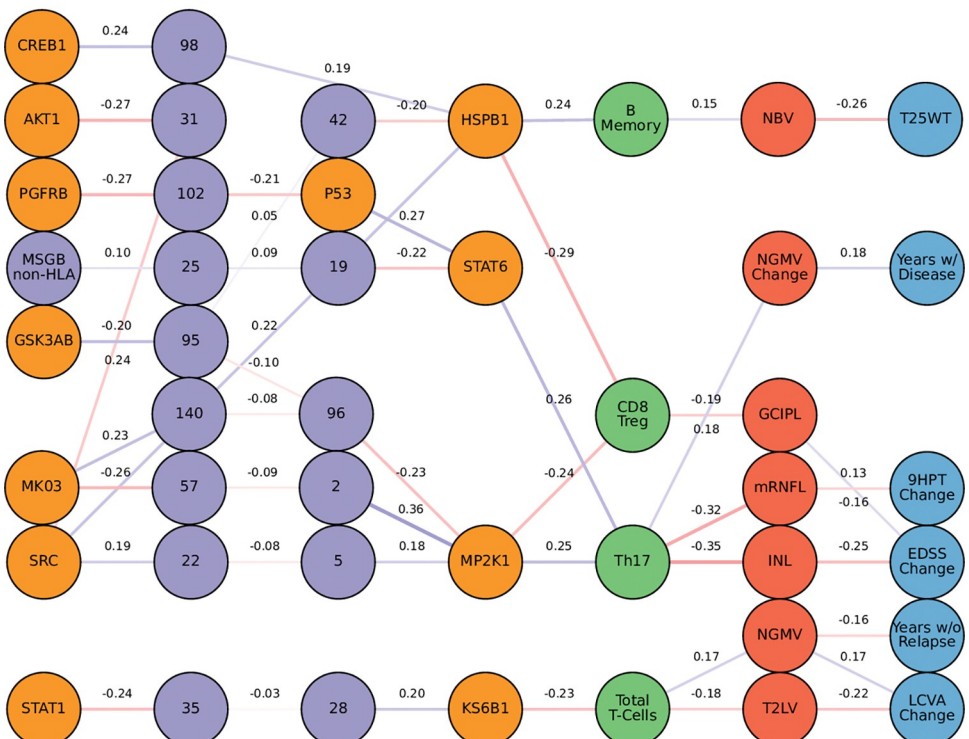

**Fig 8. The network of the top paths associated with MS.** paths staring at the genetic layer (MSGB).

MS patients and 20 HCs from the Sys4MS cohort from which frozen PBMCs were available from the baseline visit.

First, we found significant linear regression models for each of the three kinases predicting the phenotype (**Fig 10 and,** see **S6, S7, and S8 Files** for $R^2$ and p-values). In the case of GSK3AB, we found significant regression models explaining disease duration, walking speed, retina, and gray matter atrophy. For HSPB1, significant regression models were found for global disability scales such as the EDSS as well as domain specific disability scales (motor, vision, cognition), disease duration and change in gray and white matter volume. Finally, for RS6 the significant regression models also explained changes in global and motor disability (GMSSS and 9HPT) as well as retina and brain atrophy.

We then applied the single-cell data to our multilayer network and paths shown in **Fig 4**. The network was made using the significant values from the linear regressions to relate phosphoprotein-cell layer to the phenotype. With each protein (GSK3AB, HSPB1, RS6), wherever there was a significant value between a cell and phenotype, an edge was placed between the protein and cell, and another edge between the cell and the phenotype. For example, there is a significant model between the percentage of B Memory cells expressing GSK3AB and the INL change, so the two edges GSK3AB > B Memory and B Memory > INL Change are added. Edges between the imaging and clinical layers are formed indirectly, where two nodes are connected if they had at least one significant regression model with the same cell type. For example, since there are significant models between Total Treg and NBV, as well as between Total

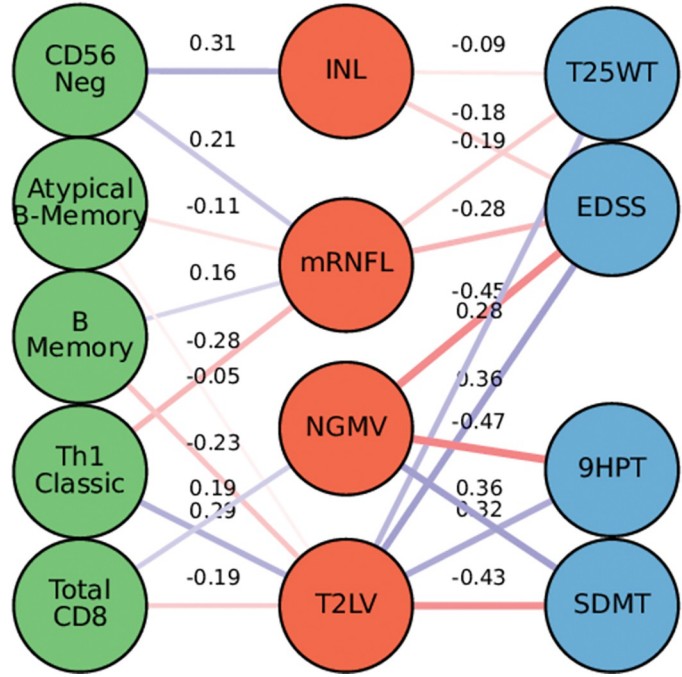

**Fig 9. The network of the top paths associated with MS.** paths starting at the cytomic layer.

Treg and EDSS, the edge NBV > EDSS is added. Next, edges between the cellular and clinical layers are removed. Finally, only the edges that are also found in the top paths from the five-layer network shown in **Fig 4** are kept. The top paths beginning with GSK3AB, HSPB1, and RS6 are listed below, and a visualization of the paths is shown in **Fig 11**.

## Comparison of paths with known biological pathways

The paths identified from the Boolean simulations were compared to known biological pathways obtained from the Uniprot database [32]. Specifically, we searched for paths that contained at least two nodes that were part of the same biological pathway. In **Fig 12** we show the overlap between the paths identified in our analysis that contain one edge validated in the single-cell analysis and the paths from UNIPROT that contain at least two nodes from UNIPROT biological pathways. We found that four out of 10 paths were present on both datasets, whereas 2 paths were present only in our single-cell analysis and 4 were only observed in the UNIPROT database. The full list of details about paths and their connections to biological pathways are given in **S9 File** and **S5 Fig**. Furthermore, one of the edges that connects the cytomics and imaging layers (Total T Cells—mRNFL) was validated by the single-cell regression analysis

In order to refine the top path found in our analysis, **Fig 13** shows Path (1) with the connections to biological pathways for the path MK03 > Total T Cells > mRNFL > T25WT, one of

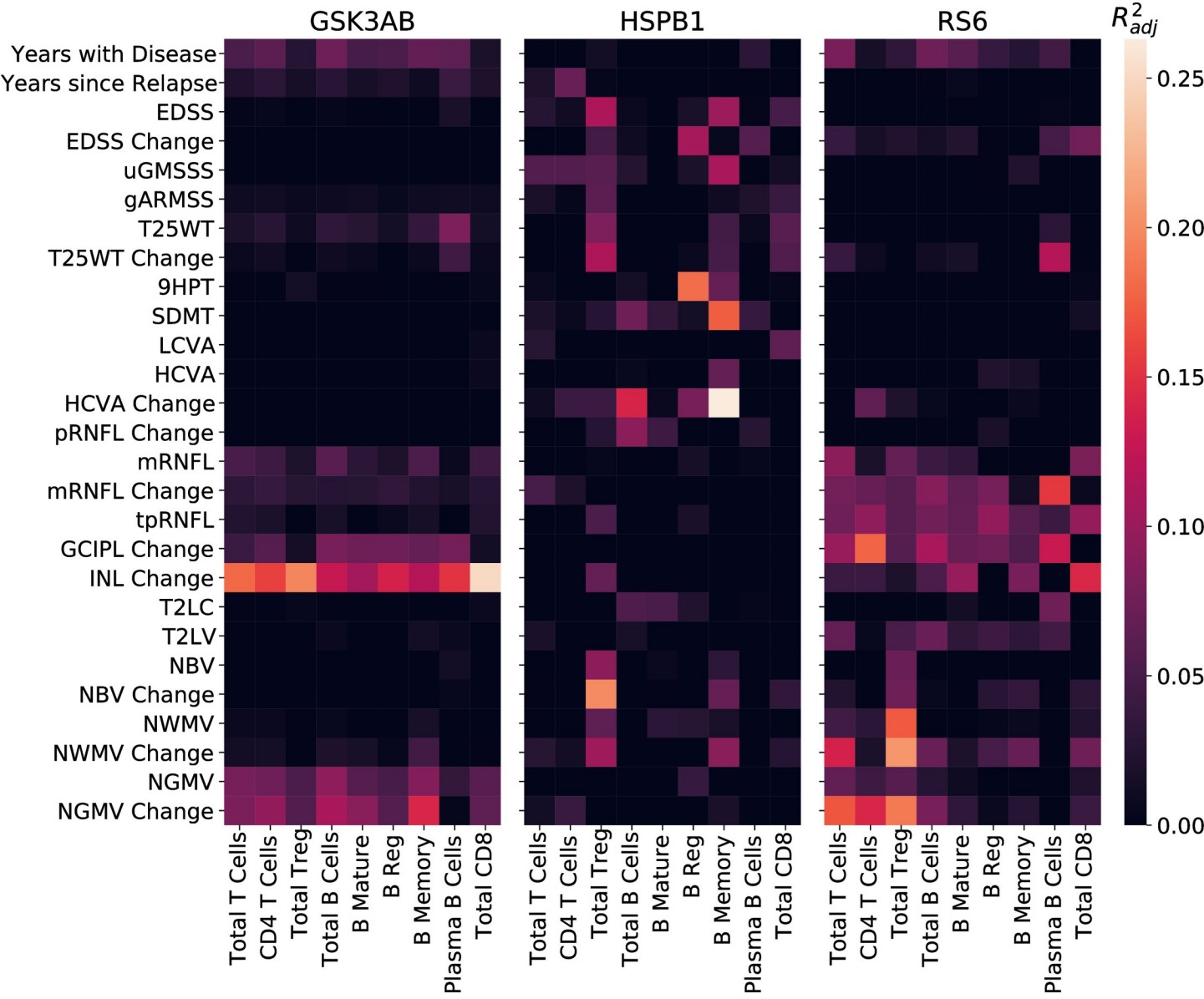

**Fig 10. Linear regression models between phosphoproteins, cell subtypes and clinical phenotype.** Linear regression analysis relating the percentage of immune cell subtypes expressing phosphorylated GSK3AB, HSBP1 or RS6 with the phenotype. The heatmap shows the adjusted $R^2$ of the significant models. EDSS: Expanded Disability Status Scale; GMSSS: Global Multiple Sclerosis Severity Score; T25WT: timed 25 feet walking test; 9HPT: nine- hole peg test; LCVA: low contrast (2.5%) visual acuity; HCVA: high contrast visual acuity; RNFL: retinal nerve fiber layer (m: macular; tp: temporal peripapillary); INL: inner nuclear layer; T2LV: T2 lesion volume; ORL: outer retinal layer; NBV: normalized brain volume; NWMV: normalized white matter volume; NGMV: normalized gray matter volume.

the paths that was identified when proteins were used as sources for the perturbations in the simulations. Many top paths contain this path within, and for this reason we postulate this path as the backbone of the MS network. There are two subpaths associated with the backbone that connect to the source MK03, one starting with GSK3AB, and the other starting with MP2K1. These subpaths contain nodes that are part of various biological pathways, including protein phosphorylation, glial cell differentiation, and regulation of stress-activated MAPK cascade. The full list of details about paths and their connections to biological pathways are given in S6 File. Furthermore, one of the edges that connects the cytomics and imaging layers (Total T Cells—mRNFL) was validated by the single-cell regression analysis.

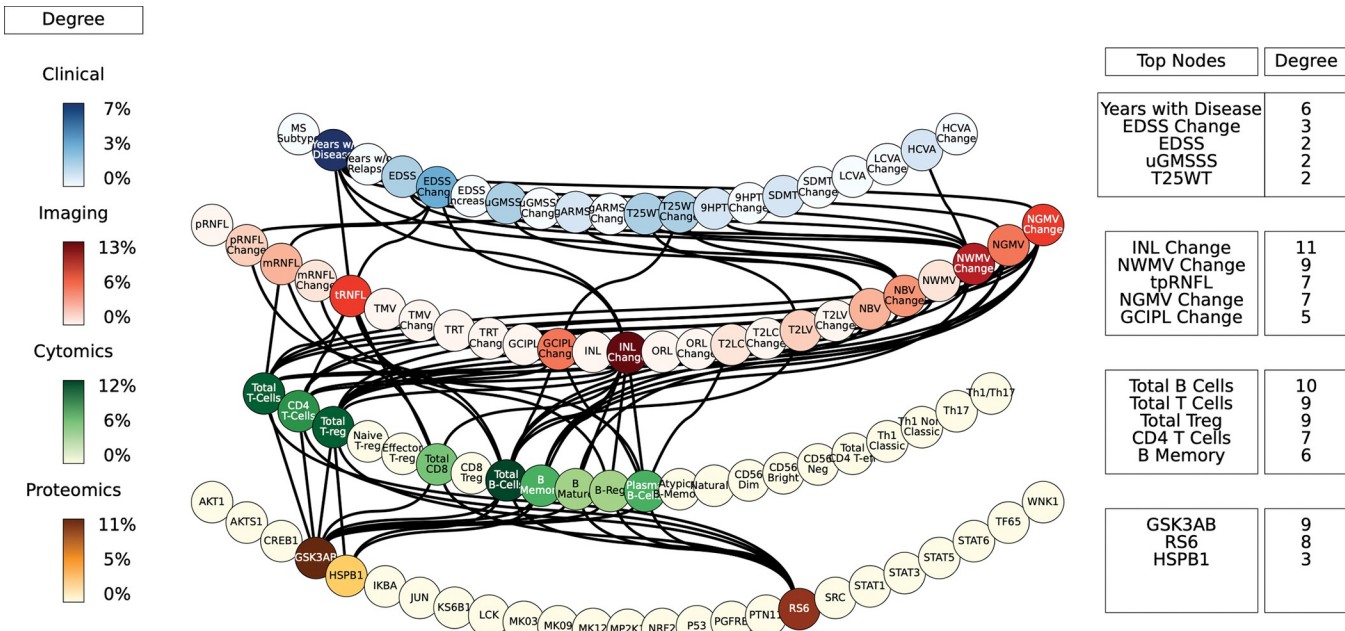

**Fig 11. Multilayer paths from single cell cytometry assays.** Each of the edges was defined using the linear regression analysis of the flow cytometry data. An edge is considered if it was part of a significant regression model and also appeared as part of a path in the original five-layer network constructed from MS patient data (from **Fig 4**). The edges are weightless, and only show if that particular edge in any of the original paths was present. https://keithtopher.github.io/fivelayer_pathways/.

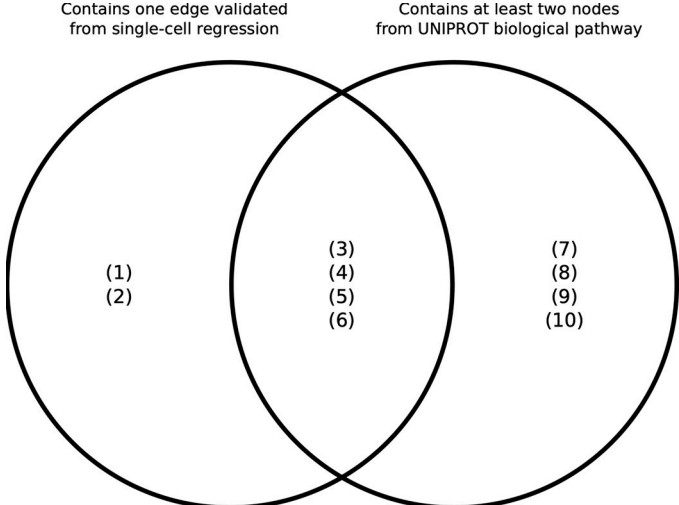

**Fig 12. Venn diagram describing the overlap between the paths identified in the single-cell analysis and the paths identified in the UNIPROT database.** (1) CD56 Neg > INL—mRNFL > EDSS—T25WT; (2) Total CD8 > NGMV— T2LV > EDSS - 9HPT–SDMT; (3) MK03 > Total T Cells > mRNFL > T25WT; (4) HSPB1 > B Memory > NBV > T25WT; (5) STAT6 > Th17 > NGMV Change > Years with Disease; (6) KS6B1—LCK > Total T Cells—Th1 Non Classic > NGMV—T2LV> LCVA Change—MSSS—Years since Relapse; (7) MP2K1—STAT6 > Th17 > mRNFL > T25WT—ARMSS (8) MP2K1—STAT6 > Th17 > INL > EDSS Change; (9) MP2K1 > CD8 Treg > GCIPL > EDSS Change; (10) Atypical B Memory–B Memory–Th1 Classic > mRNFL–T2LV > EDSS–T25WT.

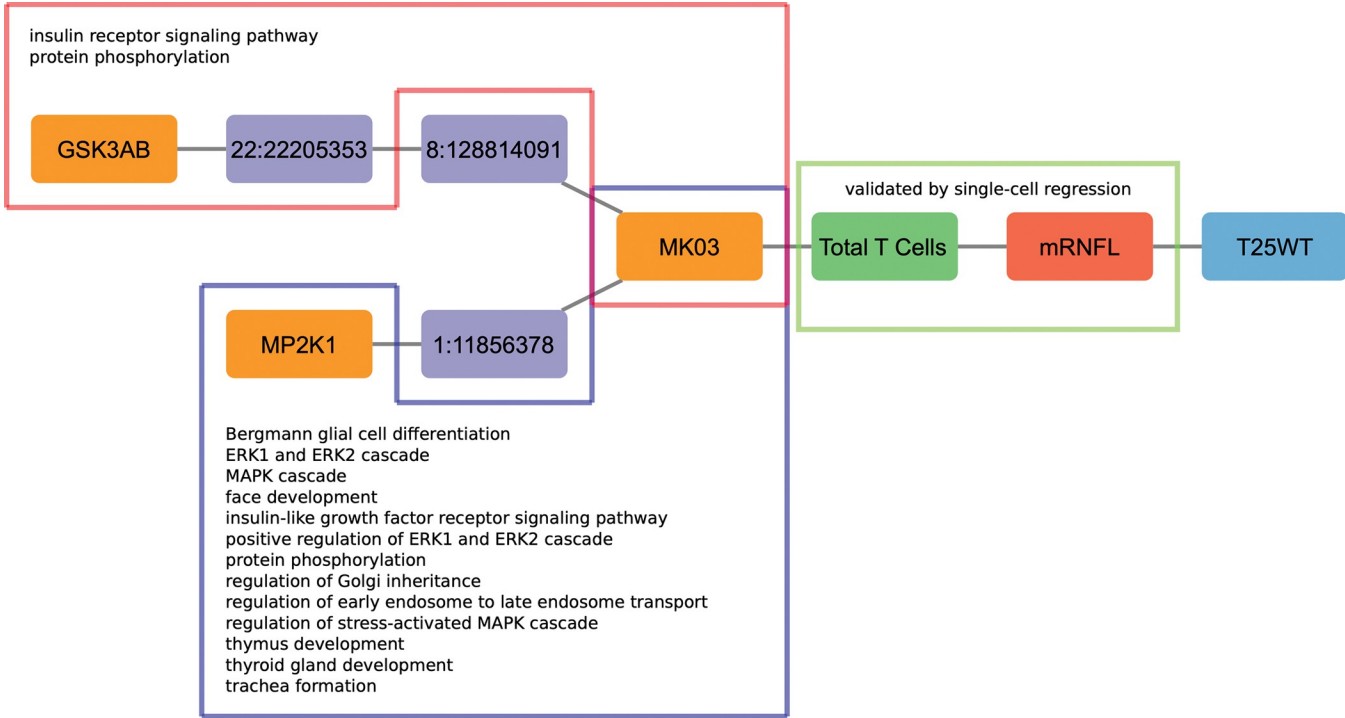

**Fig 13. Connection of the path from Boolean simulations with biological pathways (Path (1)) and validation by single-cell analysis.** Highlighted in the red and blue boxes are the nodes that are involved with the biological pathways written inside. Highlighted in the green box is the edge that was validated by single-cell regression.

## Algorithm validation in an independent diabetes' dataset

Validation of the path identification algorithm was performed using datasets from the study of a longitudinal multi-omics of host-microbe dynamics in prediabetes [2]. These datasets included six layers total: RNA-seq, cytokines, general proteins, nasal microbes, gut microbes, and clinical tests. The original data is longitudinal, where data was collected periodically from patients over the course of several years. These included healthy visits and visits where the patient had an upper respiratory infection. Samples of these visits were taken so each group (healthy and infected) contained a visit from each patient. Not all patients had a visit with an infection, so the data is skewed to healthy (n = 84) and infected (n = 25). Networks were created from each of the six datasets. Top paths were identified through Boolean simulations in the same manner as the Sys4MS datasets.

The paths identified from the Boolean simulations were compared with molecules from canonical pathways that were curated in the diabetes study [2]. These pathways include acute phase response signaling, neuroinflammation signaling, and Th1 and Th2 activation. Molecules from these pathways are present in the RNA-seq, cytokine, and protein datasets. The Boolean paths identified in this study containing at least two molecules from different layers from a given canonical pathway were considered.

By applying our algorithm to the diabetes dataset we found significant correlations between nodes of the Boolean paths with respect to the various canonical pathways. In many cases, the Pearson correlation was higher in the paths identified in the infected case compared to the healthy case. These Boolean paths contained molecules present in the following canonical pathways: 1) Acute phase response signaling; 2) Huntington's disease signaling; 3) Neuroinflammation signaling; 4) Role of JAK1 and JAK3 in gamma-c cytokine signaling; 5) Role of

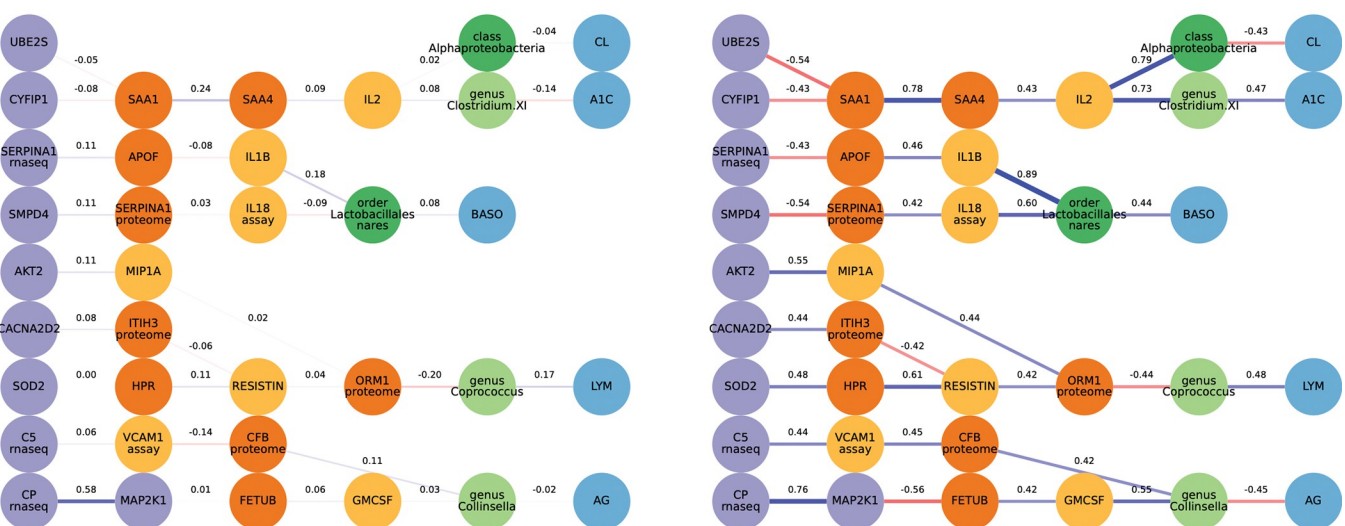

**Fig 14. Difference in Pearson correlation between healthy and infected cases.** The networks shown contain paths that were identified from the Boolean simulations in the infected network. Furthermore, each path contains at least two nodes from two different layers that are present in the acute phase response signaling biological pathway. The same paths do not necessarily appear in the healthy network, so edges with Pearson correlation are shown. There is a notable increase in the strength of the connections, both positive and negative, in the infected case.

macrophages, fibroblast, and endothelial cells in rheumatoid arthritis; 6) Role of pattern recognition receptors in recognition of bacteria and viruses; 7) Role of tissue factor in cancer; and 8) T helper cell differentiation. Visualizations of the Boolean paths that contain molecules from sample canonical pathways are available in **Fig 14**. In summary, our dynamic network analysis algorithm was able to identify the canonical pathways from the diabetes study and provide additional paths linking multiple biological layers, supporting the generalization of the approach to other datasets. Further paths can be seen at https://keithtopher.github.io/canonical_pathways/.

## Discussion

Network approaches have been very fruitful in the past at shedding light on the molecular complexity of diseases, beyond the traditional single gene and single pathway perspectives. In the traditional network paradigm, molecular components are connected according to their biological interactions, and the structure and dynamics of such interaction networks can reveal disease modules and nonlinear pathways [33]. Recently, these approaches have been extended to include multiple biological layers, such as diverse tissues with distinct protein-protein interaction networks [34], and different biological processes (membrane potential dynamics and signaling) within insulin-secreting cells [35]. Attempts have been made to construct multilayer networks for complex diseases, an approach successfully exploited in cancer research [36–41]. In this study we have applied a multilayer network analysis to integrate omics, imaging, and clinical information from patients with a complex autoimmune disease such as MS.

Our multilayer network analysis allowed us to assess the relationship between different biological scales in the disease and to identify paths linking the five layers (genomics, proteomics, cytomics, imaging and clinical) based on statistical associations. The most relevant multiscale path from our study are the one involving MK03, T cells and with retina thickness and walking speed (Path (1)) as shown in **Fig 13**.

The interaction of several phosphoproteins-cell paths and the phenotype were validated by flow cytometry studies, which were based on single cell analysis. The data from the single-cell validation concerned the amount of GSK3AB present in the cells, and one of the subpaths of Path (1) indeed contains GSK3AB. This helps in justifying the makeup of the entire path seen in Fig 8. Furthermore, nodes from Path (1) are involved in a number of biological pathways, suggesting potential mechanisms for how disruptions in this path can have a negative effect on the organism scale. The validation of Path (1) suggests a way in which processes at smaller scales (the biological pathways) manifest themselves in the larger scales (cell counts, tissue damage, and walking speed). A multi-layer network analysis is thus able to identify a differential activation of the immune system's multiple scales in MS patients that drives the phenotype.

The results from the multilevel network analysis with the omics data and phenotype data highlight the importance of considering MS as a multiscale disease, where the layers connect with varying strengths and information is filtered or strengthened across the layers [35,40]. Previous studies attempted to directly link the genomic layer with the phenotypes in many complex diseases, including MS. However, genotypes or the polygenic risk scores alone have a limited ability to predict either the cell variability or the phenotype [42, 43]. Other genomic information such as DNA sequencing, epigenetics and RNA expression, or more global approaches is likely needed for a more thorough analysis in multiscale complex diseases.

The kinases studied are part of pathways previously described as associated with MS (reviewed in [9]). MK03 (aka MAPK3/ERK1) mediates diverse biological functions such as cell growth, adhesion, survival and differentiation through the regulation of transcription, translation, cytoskeletal rearrangements [44] and its expression has been associated with MS as an expression quantitative trait loci [45]. [44] MP2K1 was the kinase showing the strongest association with the presence of MS in our previous study [26] and is a master regulator of the immune response. We and others have previously described increased GSK3AB expression or phosphorylation levels in patients with MS [26,46,47]. GSK3 plays key roles in Th1 cell activation as well as in microglia modulation, in addition to its effects on neuronal survival and functioning [46]. HSBP1 (aka HSP27) is a stress protein that in addition to its chaperone activity, is critical for apoptosis signaling pathways within the mitochondria, inhibiting the Apaf complex [48]. Indeed, HSBP1 has been found to be increased during MS relapses [49]. RS6 is a MAPKinase that is modulated by extracellular signal-regulated kinase (ERK) and activates serum glucocorticoid kinase 3 (SGK3), nuclear factor kappa-light-chain-enhancer of activated B cells (NfKB), mammalian target of rapamycin (mTOR) and other pathways modulating cell growth and differentiation. Inhibition of ERK and RS6 in models of MS reduces proliferative response, phagocytic properties, and synthesis of proinflammatory mediators induced by the addition of inflammatory stimuli to microglia [50]. Regarding the immune cell subtypes highlighted in our analysis, our previous analyses of the Sys4MS dataset support the results of the current network analysis that confirms the prominent role of B cells in MS [24]. Such results agree with our previous analysis of phosphoproteins and immune cell subtypes in another dataset of MS patients showing the preferential involvement of B cells [26]. Many pieces of evidence have confirmed a remarkable role for B cells in MS [51], probably driven by the latent infection of the Epstein-Barr virus that produces immune response dysregulation or molecular mimicry with CNS proteins like GlialCAM [16, 52]. In addition, CD8 cells are the most abundant cell type in the brain infiltrates [11]. Finally, a recent study in twins discordant for MS is also providing new endorsement of the role of helper CD4 cells [12].

The data provided by the Sys4MS cohort was rich in the wide range of scales it covered. However, several limitations were encountered with both the data and analysis. Although the sample size of the cohort was enough to identify significant correlations, the sample sizes were smaller for some specific omics (proteomics and cytomics), although bigger than n-of-1

studies commonly used for deep phenotyping [40]. The limited sample size may have affected both the networks constructed as well as the statistical tests conducted with the paths or for the analysis stratifying by each of the therapies. Furthermore, the omics dataset collected were cross-sectional, whereas the imaging and clinical data were longitudinal. Longitudinal data from all five layers and deep phenotyping would greatly benefit future studies. Second, we were unable to collect cells from the CNS for omics analysis and for this reason we relied on imaging data for modeling CNS damage introducing additional noise. Another concern is the validation of the paths because deep phenotyped MS cohorts are not available. A wealth of MS patient data from other studies is available with genomics, imaging, and clinical phenotype (through the IMSGC and MultipleMS consortia). However, proteomics, cytomics or other types of omics data is usually lacking, which limits conducting validation in independent datasets. Further limitations relate to the omics experiments themselves. For this reason we conducted a validation of our algorithm in an independent multi-omics dataset from patients with diabetes. Both the protein and cell analyses were conducted using PBMCs, rather than in immune cells from the central nervous system. Also, the protein analysis was not performed at the single cell level but in bulk PBMCs in the overall cohort. Therefore, single-cell dynamics were not captured in the first experiment. However, flow cytometry analysis performed for the validation study provided single-cell information which supports the validity of the findings. Additionally, limitations were also partially balanced by using a hypothesis-driven design that included kinases and cells previously described as differentially activated in MS.

In summary, this study examined the functional connections among various scales of biological data of a complex disease with a complex genetic basis, namely MS. Our multilayer networks support that information flow across scales. Indeed, we have identified a path involving the kinase MK03, T Cells and retina thickness that predicts disability (Path 1). This highlights the importance of the molecular and cellular scales when considering explaining the phenotypes of complex diseases. Indeed, these paths could be the target of a future treatment of personalized medicine in MS. This could also be transferable to other autoimmune disorders, commonly sharing disease underlying mechanisms.

## Methods

### Ethical statement

The Sys4MS project was approved by the Institutional Review Boards at each participating institution: Hospital Clinic of the University of Barcelona, IRCCS Ospedale Policlinico San Martino IRCCS, Oslo University Hospital, and Charité—Universitätsmedizin Berlin University. The Barcelona MS cohort study was approved by The Ethic Committee of Clinical Research, Hospital Clinic Barcelona. Patients were invited to participate by their neurologists, and they provided signed informed consent prior to their enrollment in the study. De-identified data were collected in a REDCap database at the Barcelona center.

### Patients

**Sys4MS cohort.** We recruited a cohort of 328 consecutive MS patients according to 2010 McDonald criteria [53] and 90 healthy controls (HC) at the four academic centers: Hospital Clinic, University of Barcelona, Spain (n = 93); Ospedale Policlinico San Martino, Genova, Italy (n = 110); Charité—Universitätsmedizin Berlin, Germany (n = 94); and the Department of Neurology, Oslo University Hospital, Norway (n = 121) as described before [24]. 99% of the cohort was Caucasian, which is reasonable considering MS predominantly affects Caucasians compared to other ethnicities and the referral populations from our centers. We collected clinical information (demographics, relapses, disability scales, and use of disease-modifying

drugs), and imaging data (brain MRI and OCT), and obtained blood samples at the same visit. The imaging data is aimed to capture the damage of the CNS, whereas the immune cells data (genomics, proteomics, and cytomics) aims to capture the dynamics of the immune system during the autoimmune response. Patients were required to be stable in their DMD use over the preceding six months. Patients were followed for two years, and the same clinical, disability scales, and imaging data (brain MRI and OCT) were collected at the 2-year follow-up visit.

**Clinical variables.** Each patient was assessed on the following disability scales at baseline and follow-up: the Expanded Disability Status Scale (EDSS); timed 25 feet walking test (T25WT), nine-hole peg test (9HPT), the Symbol Digit Modality Test (SDMT), 2.5% Sloan low contrast visual acuity (SL25), and high contrast vision (HCVA, using EDTRS charts and a logMar transformation). We calculated the MS Severity Score (MSSS) and the age-related MS Severity Score (ARMSS).

The ARMSS was used for dividing the cohort based on disease severity using the tertile distribution (first tertile were mild MS, the second tertile was excluded and the third tertile were defined as severe MS). Change in the disability scales and 2-year follow-up visit was calculated as the difference (delta) between the two visits. EDSS changes were confirmed in a clinical visit 6 months before the study follow-up visit. At each visit, we collected the information regarding the patients' DMD use, including low-efficacy therapy: interferon-beta, glatiramer acetate, and teriflunomide; or mid to high-efficacy therapy: fingolimod, dimethyl-fumarate, natalizumab, or other monoclonal antibodies (alemtuzumab, rituximab, daclizumab, and ocrelizumab).

**Imaging.** MRI studies were performed on a 3-Tesla scanner at each center using a standard operating procedure (SOP) to optimize the volumetric analysis. We used the 3-dimensional (3D) isotropic T1-weighted magnetization-prepared rapid gradient echo (T1-MPRAGE) (resolution: 1 x 1 x 1 mm$^3$), and 3D T2-fluid-attenuated inversion recovery (T2-FLAIR) images with the same resolution to quantify changes in brain volume. Presence of contrast-enhancing lesions, T2 lesion volume, new or enlarging T2 lesions, and volumetric analysis were done at the Berlin center as previously described [54]. Retinal OCT scans were performed using the Spectralis device in three centers and the Nidek device at Oslo center. A single grader at the reading center in Berlin performed intra-retinal layer segmentation using Orion software (Voxeleron Inc, Berkeley, US) to quantify the macular ganglion cell plus inner plexiform layer (GCIPL) and the macular inner nuclear layer thicknesses (μm) in the 6 mm ring area as previously described [55].

**Flow cytometry.** The original cytometry data was obtained on fresh peripheral blood mononuclear cells (PBMCs) using 17 antibodies that covered 22 cell subpopulations of T, B and NK cells as described in detail elsewhere [24]. The following cell populations were studied: T cells: CD3+, CD3+CD4+, CD3+CD8+; B cells: CD19+; and NK cells: CD3-CD14-CD56+, as well as the specific subpopulations: Effector cells: Th1 classic: CD3+CD4+CxCR3+-CCR6-CD161-; Th17: CD3+CD4+CxCR3+CCR6-CD161+CCR4+; Th1/17: CD3+CD4+-CCR6-CD161+CxCR3highCCR4low; Regulatory T cells: CD3+CD4+: Treg CD25+CD127-, T naive CD45RA+CD25low; CD3+CD8+: T reg CD28- and T naive CD28-CD45RA+; B cells: B memory: CD19+CD14-CD24+CD38-; B mature: CD19+CD14-CD24+CD38low; B regulatory: CD19+CD24highCD38high and NK cells: Effector: CD3-CD14-CD56dim: Regulatory: CD3-CDCD56bright (reg). For validation assays, PBMC in triplicate tubes were stained with BV510-conjugated anti-CD3 (Clone OKT3, Catalog # 317332, BioLegend)), APC Cy7-conjugated anti-CD4 (Clone SK3, catalog #344616, BioLegend), BV421-conjugated anti-CD25 (Clone BC96, catalog # 302630, BioLegend), AF700-conjugated anti-CD127 (Clone A019D5, catalog # 351344, BioLegend), PE Cy7-conjugated anti-CD19 (Clone HIB19, catalog # 302215, BioLegend), PE-conjugated anti-CD24 (Clone ML5, catalog # 311105, BioLegend) and PE/Dazzle594-conjugated anti-CD38 (Clone HB-7, catalog # 356630, BioLegend) antibodies in

solution for 30 min at 4˚ C and washed twice with PBS. The cells were then fixed and permeabilized with Cytofix/Citoperm (BD Bioscience), according to the manufacturer's instructions. For intra-cellular staining, the cells were blocked with 5% normal goat serum for 20 min on ice to prevent non-specific binding of the antibodies, and stained for total and relevant phospho-proteins with the following antibodies in one of the three tubes: Tube 1: mouse monoclonal anti-human RPS6 (Clone 522731, catalog # MAB5436, R&D Systems) and rabbit polyclonal anti-human Phospho-RPS6 (Catalog # AF3918, R&D Systems); Tube 2: rat monoclonal anti-human GSK-3B(Clone 272536, catalog # MAB2506, R&D Systems) and rabbit polyclonal anti-human Phospho-GSK-3BCatalog # AF1590, R&D Systems); and Tube 3: mouse monoclonal anti-human HSP27 (Clone G31, catalog # 2402; Cell Signaling Technology) and rabbit polyclonal anti-human Phospho-HSP27 (Catalog #AF2314, R&D Systems) antibodies. All primary antibodies were used at a concentration of 5 μg per 1 x 106 cells. The cells were then washed twice and incubated on ice for 15–20 min with the appropriate fluorescent-conjugated secondary antibodies, Alexa Fluor 488-conjugated goat anti-rabbit IgG (Catalog # A-11070, Invitrogen; 1:100 dilution), APC-conjugated goat anti-mouse IgG (Catalog # 405308, BioLegend; 1:100 dilution), or APC-conjugated goat anti-rat IgG (Catalog # 405407, BioLegend; 1:100 dilution), in 5% normal goat serum. The cells were then washed twice, resuspended in assay buffer, and analyzed on a Beckman Coulter Navios flow cytometer. Analysis was performed using Kaluza software. Phosphorylation levels were defined in terms of mean fluorescence intensity (MFI) of phosphorylated protein over MFI of total protein. A representative cytometry plot for each of the three phosphoproteins is shown in **S4 Fig**. Gating strategy and representative cytometry plots for showing the cell sorting and signal intensity for phospho-GSK3Ab, phospho-HSBP1 and phospho-SR6 assays.

## Genotyping

Genotyping of the samples was performed by FIMM Genomics (University of Helsinki, Finland) on the Illumina HumanOmniExpress-24 v1.2 array (713,599 genotypes from 396 samples). SNPs imputation was conducted against the 1000-genomes reference (quality of imputation $r^2 > 0.5$; 6,817,000 genotypes for 396 samples), which allowed to extract MS-associated SNPs (152 out of 200 known MS-associated SNPs available and 17 out of 31 known MS-associated HLA alleles available (HLA*IMP program)) as described elsewhere [56]. The MS Genetic Burden Score (MSGB) for the HLA and non-HLA alleles and their combination was calculated as described previously [27]. Briefly, the MSGB is computed based on a weighted scoring algorithm using one SNP per MS associated genomic region as found by trend-test association (meta-) analysis. This statistic is an extension of the log additive model, termed "Clinical Genetic Score", with weights given to each SNP based on its effect size as reported in the literature. The MSGB is obtained by summing the number of independently associated MS risk alleles weighted by their beta coefficients, obtained from a large GWAS meta- analysis, at 177 (of 200) non-MHC (major histocompatibility complex) loci and 18 (of 32) MHC variants, which includes the HLA-DRB1*15:01-tagging single-nucleotide polymorphism (SNP) rs3135388. Considering the binary nature of the SNPs themselves, we used the MSGB scores, which allow continuous distributions for genetic information, allowing the planned network analysis.

## Expanded genetic network including regulatory network information

The SNPs were mapped with their nearest gene by the IMSGC consortium [8], and a network was constructed using data from the Gene Regulatory Network Database (GRNdb) [29]. The database provides networks of transcription factors (TFs) from various cell types in the human

body. The gene regulatory network (GRN) within PBMCs was used containing 12,878 genes, of which we only considered the subset of genes that were mapped to the SNPs from our study. Taking a subset in this way causes some of the regulatory information to be lost, such as two genes that are regulated by the same TF. There is still a relationship between two such genes, although indirect. To include this information in the network of MS genes, an edge was added between two genes that share a transcription factor. The underlying GRN provides us a reference network of how the genes interact in the healthy state. The allelic (SNP) information alone is binary and does not change throughout time, limiting the analysis. Considering the underlying GRN strengthens the analysis by allowing us to integrate the MS-related SNPs into a model that considers dynamic behavior. Similar approaches have been seen in previous studies, where a previous knowledge network of protein signaling interactions is used as a base for logic modeling that incorporates new data to determine the differences between healthy and diseased states [28]. The mapping of the GRN, along with the MSGBs mentioned before, allows the genetic data to be better integrated with the remaining layers.

Once the GRN of MS genes was obtained, each gene was then replaced with its corresponding SNP. This is not a one-to-one mapping, as there are some SNPs that are mapped 13 to the same gene. In this case, edges are placed among all SNPs that share a gene. This allows the GRN to be compared with the other layers in the combined network. Finally, only edges that appear in the original network of SNPs connected with Pearson correlation are kept, and their weights are used in the GRN. Details of these networks can be found in https://keithtopher.github.io/networks/#/.

## XMAP Phosphoproteomics

Phosphoprotein levels were quantified using xMAP assays performed blindly at ProtAtOnce (Athens, Greece) as described previously [26,28]. We analyzed a set of kinases associated with MS(9) which provides an adequate signal to noise ratio and test-retest reproducibility: AKT1, AKTS1, CREB1, GSK3AB, HSPB1, IKBA, JUN, KS6B1, LCK, MK12, MK03/01, MK09, MP2K1, NRF2, P53, PGFRB, PTN11, RS6, SRC, STAT1, STAT3, STAT5, STAT6, TF65, WNK1. Phosphoprotein data was normalized after the measurements were taken as described elsewhere [28].

## Data processing

The omics and clinical datasets were ultimately used to build the multilayer network, where each dataset represents a layer in the network. The data were examined to handle missing values, identify which patients have data from which layers, as well as divided into groups based on gender, disease severity, medication, etc. No imputation was used in this study. Patients were divided into mild and severe groups according to the tertiles of their age-related multiple sclerosis severity (ARMSS) score. Patients in the lower 40th percentile were classified as mild, and those in the upper 40th percentile classified as severe. This division between mild and severe inherently stratifies the patients into those who receive low-efficacy and high-efficacy treatments, respectively. The 2-year follow-up data from the clinical and imaging layers were used to calculate the change from baseline, and these changes were added as new variables.

## Multilayer network construction

For the structural/topological network analysis we used mutual information for defining the edges due to the ability to identify linear and non-linear interactions. For this dynamic network analysis we used Pearson correlation for identifying the layers because even if it only identifies linear correlation, it provides negative and positive correlations which are required

for defining the AON and OR gates of the Boolean analysis. Individual structural networks were constructed from the five layers by computing mutual information between nodes within each layer, due to the inherent nonlinear nature of biological processes. First, the networks within an individual layer were constructed, and then the networks across layers (see **Fig 2** for details on degree distribution for each layer). This step was done separately for two reasons: first to highlight the inherent differences (including biological scale) among the various layers, and second to utilize the maximum number of subjects available for each dataset. This is because not all subjects have data for both cytomics and proteomics.

Once individual layer networks were constructed, the features between layers were connected together, again with mutual information. Not all layers are interconnected, however, due to a predetermined hierarchy applied to the system (see **Fig 1G**). Ultimately, this produced a network of five connected layers, where each layer contains features from each of the five original datasets. A pipeline for the construction of the networks is shown in **Fig 1**. A second type of network was constructed using all five datasets, this time using linear correlation to define the edges, and such a network was later used in the path analysis.

## Calculation of correlation for edges

The method to calculate the edge weights in our networks was adopted from the ARACNE method [57] and simplified. The networks were constructed using mutual information, using the traditional binning method to calculate the mutual information pairwise between all the elements within individual layers and later between layers [58]. The data for a given element are split into 10 equally spaced bins, and the probability of falling within a certain bin is calculated for each element individually as well as the joint probability for a two-point coordinate falling within a certain two-dimensional 1/10 by 1/10 size bin. The formula for the mutual information between two variables X and Y is

$$I(X, Y) \ = \ \Sigma_{ij} p(i,j) log \frac{p(i,j)}{p_x(i) p_y(j)}$$

where px(i) and py(j) are the marginal probabilities for variables X and Y, respectively, and p(i, j) is the joint probability between X and Y. The python package *scikit-learn* [59] was used for the mutual information calculation.

Once the mutual information value is calculated, a threshold is needed to determine if there is indeed a correlation between the two elements. Random permutations over subjects are performed separately for both variables, and the mutual information is calculated over the permuted data. This process is repeated 1000 times, and a distribution is obtained of random mutual information values (surrogates). The mutual information value obtained from the original data is compared to the distribution of random values to determine if it is significantly higher than the distribution. The distribution is treated as Gaussian, and the original mutual information value is considered significant if it passes a z-test with p-value lower than $p = 0.05$. Edges are placed between all significant pairs. Weights are assigned using the normalized value of mutual information, which falls between 0 (no correlation) and 1 (perfect correlation).

The combined network (later used for the path analysis) was constructed using Pearson correlation. The Pearson correlation coefficient was calculated pairwise between each of the elements included in the two datasets, using the python package *scipy* [60]. An edge was defined if the p-value associated with the correlation was lower than $p = 0.05$. Next, the value of the Pearson correlation itself was used as the weight of the edge, giving a weight that falls between -1 (perfect negative correlation) and 1 (perfect positive correlation).

### Path identification via Boolean modeling

The method of path identification was inspired by Domedel et al. [31], which studied the information processing within a cell signaling network by utilizing Boolean modeling. The combined five-layer network was constructed using Pearson correlation, and information flow across it was analyzed using Boolean simulations. This is done to examine how perturbing the network affects nodes within the various layers, especially those representing the phenotype. The genomics network in this case was modified further, utilizing information about regulatory interactions from the Gene Regulatory Network Database [29], between the genes that are mapped to the SNPs (described further above). The exact chemical reactions between proteins and cells are ignored, giving a qualitative description of the system [30]. The goal of this step is to identify differences in paths responsible for triggering immune responses in healthy subjects compared to MS patients.

For simplicity, each element in the network (from one of the five layers) is considered to be in one of two states: active/inactive. For example, this represents high/low levels of phosphorylation for proteins. The Boolean simulation begins in a random state where each element has a 50% probability of starting as active or inactive. At each step, the elements' activation states are updated based on the sum of the states of their neighbors. The nature of the connections between elements is key, as they have either activating (positive) or inhibitory (negative) relationships. For a given node, each neighbor contributes a score based on the weight and the sign of the connection of the corresponding Pearson correlation. The total sum of the weights of the neighbors determines whether the node will be active or inactive on the next iteration.

As an example, consider the protein GSK3AB (inactive) with neighbors HSPB1 (active) and IKBA (active), as seen in **Fig 15**. Let's say there is a positive connection between GSK3AB and HSPB1 with a weight of 0.8, and a negative connection between GSK3AB and IKBA with a weight of 0.5. Since HSPB1 is active and has a positive relationship with GSK3AB, it contributes a score of +0.8 to change GSK3AB to the active state. Since IKBA is active and has a

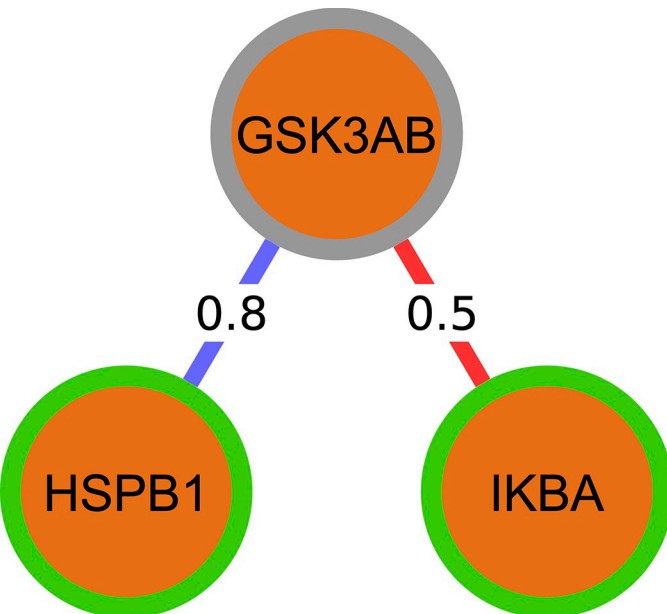

**Fig 15. Depiction of summing weights to determine next activation state in Boolean simulations.** A green border represents an active node, and a gray border represents an inactive one.

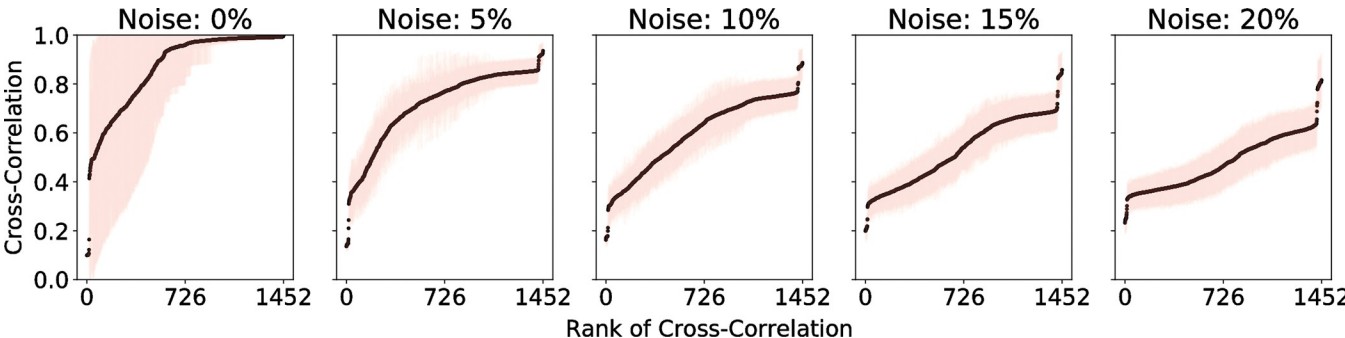

**Fig 16. Effect of noise in Boolean simulations on the cross-correlation coefficient of the signals between nodes in the combined network.** With 0% noise, a majority of the cross-correlation values are nearly 1, which does not allow the node pairs to be easily ranked based on the strength of their connections. With 5% noise, there is more deviation in the cross-correlation values, which allows the paths between a chosen source and target to be more easily identified.

negative relationship with GSK3AB, it contributes a score of -0.5 to GSK3AB inactive. Overall, we have a score of +0.3, so GSK3AB becomes active.

Each step of the simulation was run in this manner and continued for 100 steps. One of the MSGB scores, proteins or cells was chosen as the input, where it was manually flipped between active and inactive states with a defined period (in this case 10 iterations active, then 10 iterations inactive). This was done to examine how perturbations in the input node travel through the network and ultimately affect a given phenotype (output). The perturbations themselves represent changes between low to high values in the distribution for a given MSGB scores, protein, or cell. For the MSGB non-HLA score, the perturbations flip the value between high and low genetic risk. For a protein such as GSK3AB, the values flip between low and high phosphorylation. Finally for a cell such as B Memory, the values alternate between high and low cell counts.

Noise was also added to the system, where each element has a set probability of changing its state at each iteration. The effect of noise is illustrated in **Fig 16**. This addition of noise reflects the inherent stochasticity in biological systems as well as prevents the simulations from simply settling directly into a fixed state. The noise was chosen to be 5% because this allows greater differences for the cross-correlation of the signals between nodes as shown in **Fig 16**. With no noise at all, many of the nodes remain either active or inactive for the majority of the simulation. This causes the cross-correlations to be too high between nodes, and the subtle differences in the strength of the connections is not seen.

Once the simulations were run, the temporal cross-correlation function was calculated between all pairs of nodes. The cross-correlation is a measure of similarity classically used in signal processing and is the same used in [31]. The maximum cross-correlation (which could occur at a non-zero lag time) was determined, and its inverse is placed as a weight on the edges of the existing network, in such a way that a high correlation would correspond in this case to a low weight. In case there was no edge in the original network, no edge is defined in the new network either. A cell type or phenotype is selected as a target (output), and the most efficient paths are identified between it and the fixed source (input). An "efficient" path is defined as one in which the total sum of the weights (inverse maximum cross-correlations) of the edges connecting the source and target (called a path score) is lower than the rest. This definition favors both low number of steps and high cross-correlations between nodes within a path. A shortest path algorithm developed by Yen et al [61] was used, which gives precedence to the lowest path scores.

Simulations were conducted between every possible pair of inputs (MSGB, proteins, or cells) and outputs (cells or phenotypes). Overall, the simulations aim to reveal how information flows through the entire networks, providing insight on underlying pathology in MS. This

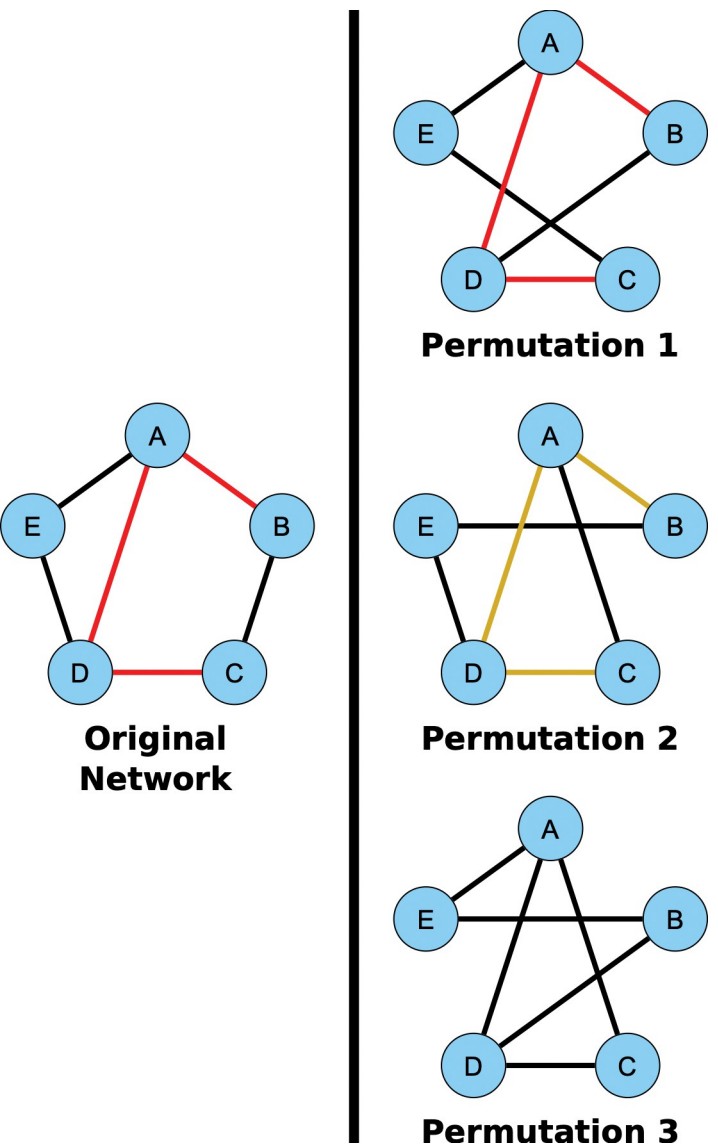

**Fig 17. Network permutation for negative controls of paths.** The five-layer network built using Pearson correlation is used as the base network. For each of the 100 repetitions, the network was permuted by swapping the edges between pairs of nodes. In permutation 1, the edge between B and C was swapped with the edge between D and E. In the permutation 2, the edge between A and E was swapped with the edge between B and C. In permutation 3, first the edge swap from the top network was applied, followed by the edge swap from the middle network. In each case, the edge swap can only be done if it does not result in two edges between the same pair of nodes. Making the permutation in this way keeps the original degree distribution of the network. The weights for each of the edges are permuted as well. This edge swapping technique is applied 10 times for each edge in the original network. After they are permuted, the top paths for each network are identified in the same manner as before. There are three possibilities for considering whether the paths from the original network appear in the paths from the permuted networks. In permutation 1, the path exists in the permuted network and furthermore was identified as a top path. In permutation 2, the original path does exist in the permuted network but was not identified as a top path. In permutation 3, the original path doesn't exist in the permuted network at all.

provides useful biological information, as differences in paths can be accessed between various subsets of patients (mild, severe, progressive MS, relapse-remitting MS, untreated, low-efficacy, and high-efficacy treatments). The algorithm for performing the Boolean simulations and the path identification is represented schematically in **Fig 2**.

In order to test the consistency of the results, we ran 100 simulations for each source, then these 100 simulations were used to calculate the cross-correlation between proteins/cells to identify the paths. We applied a jackknife resampling 10 times, first taking 90 random samples, then 80 random samples. In both cases, 9 out of 10 paths on average were identical over all protein sources and cell targets. Also, as stated in the main text, negative controls were considered by permuting the network before running the Boolean simulations. An illustration of the process for permuting the networks and identifying their corresponding paths is shown in **Fig 17**.

## Combinatorial analysis

All possible combinations of source sources (MSGB scores, proteins, cells) and targets (cells, imaging and clinical phenotype) were used to identify top paths. The simulations were run with each protein as a source, where it remained active for 10 steps, then inactive for 10 steps. After the simulations were run for each source, and the cross-correlation values were calculated, each cell type was selected to be the endpoint for the path finding algorithm. This was performed as a screening process to create an ensemble of paths for each source/target pair. Their significance in the phenotype was assessed next.

## Statistical analysis

The study was designed with a 1:4 ratio controls vs MS patients are based in the following reasoning: 1) the goal was the prediction of the phenotype and for such analysis only MS cases will be used; 2) controls were only used for the logistic regression comparing the diagnosis; 3) MS is heterogenous and for this reason it was expected to perform comparisons between subgroups based on disease subtype and therapy, requiring a bigger sample size for the MS group. For this reason, we designed a 4:1 ratio. Controls were collected in equal proportion from all participant centers in order to avoid center bias.

Descriptive statistics, normal distribution assessment, and class comparison analysis was performed for the five layers. The Mann-Whitney test was used due to non-normal distributions being present in both datasets. Mutual information was used in constructing the topological networks for all five layers.

## Network statistics

Network metrics were calculated from the networks constructed using mutual information. including average degree and density. The clinical and imaging datasets lack information from

**Table 3. Average degree of individual networks constructed using mutual information to define edges.** These degrees do not consider the connections among layers. The superscripts (a,b) represent cases where there was a significant change when comparing degree distributions. The Mann-Whitney test was used for all pairings, due to the non-normality of the degree distributions.

| | Average Degree | | | | | | |
|---|---|---|---|---|---|---|---|
| | Healthy | All Patients | Patients w/o Fingo | Mild Patients | Mild w/o Fingo | Severe Patients | Severe w/o Fingo |
| **Clinical** | - | 11.2 | 11.4 | 8.0 | 8.1 | 7.8 | 7.2 |
| **Imaging** | - | 6.3 | 6.3 | 4.6 | 4.6 | 3.5 | 3.4 |
| **Cytomics** | 2.5[a] | 8.8 | 6.0[b] | 5.8 | 3.8[b] | 5.7 | 3.6[b] |
| **Proteomics** | 5.0[a] | 8.0 | 7.5 | 3.9 | 2.9 | 5.5 | 5.8 |
| **Genomics** | 12.2[a] | 13.3 | 12.6[b] | 12.2 | 13.0[b] | 12.4 | 12.3 |

[a] *Significant increase (p-val < 0.05) in degree between healthy controls and all patients.*

[b] *Significant decrease (p-val < 0.05) in degree between all patients in a given subset (all, mild, or severe) and those not treated with Fingolimod.*

healthy controls, so networks were not constructed in these cases. The average degree is given for each individual layer for healthy controls and MS patients, including those who are not treated with fingolimod (Table 3). Considering the omics datasets, all three of cytomics, proteomics, and genomics saw a significant increase in degree from the healthy network to all patient network at the 5% significance level. When comparing groups of patients treated with any medication versus groups excluding the patients treated with Fingolimod (a high-efficacy treatment with notable effects on cell counts in the immune system[3]), the cytomics networks saw decreases in degree in every case, and the genomics saw decreases for all patient and mild patient networks.

## Supporting information

**S1 Methods. Supplementary methods.**
(DOCX)

**S1 Fig. Pipeline for network build-up for the topological and dynamic networks.**
(PDF)

**S2 Fig. Comparing the network topology of various subsets of patients from the original datasets.**
(PDF)

**S3 Fig. Comparison of paths identified for the various subsets of patients from the original datasets.**
(PDF)

**S4 Fig. Cytometry plots for the expression of phosphoGSK3AB, phosphoHSBP1 and phosphor-RS6 in immune cell subpopulations.**
(PDF)

**S5 Fig. Overlap between the paths identified in the single-cell analysis and the paths identified in the UNIPROT database.**
(PDF)

**S1 File. Variables.**
(XLSX)

**S2 File. Pathways.**
(XLSX)

**S3 File. Pathways negative controls thresholds.**
(XLSX)

**S4 File. Pathways negative controls alphas.**
(XLSX)

**S5 File. Pathways path score.**
(XLSX)

**S6 File. Linear regressions for RS6 node.**
(XLSX)

**S7 File. Linear regression for HSPB1 node.**
(XLSX)

**S8 File. Linear regressions for GSK3AB node.**
(XLSX)

**S9 File. Pathways validation for canonical pathway.**
(XLSX)

## Acknowledgments

We would like to thank the MS society of Norway and Italy and the GAEM foundation from Barcelona, Spain for the feedback provided to this project. We would also like to thank Dr David Gomez from the Department of Pediatric Neurology, Hospital Vall d'Hebron, Barcelona, Spain, for the review of the linear regression models. We also thank Dr. Ina Brorson for imputation of the genetic data and the research optometrists in prof. Liv Drolsums lab at Oslo University Hospital for help with the follow-up visual examinations and OCT scans and Fernanda Kropf and Ingrid Mo for technical assistance in preparation of biological samples.

## Author Contributions

**Conceptualization:** Keith E. Kennedy, Antonio Uccelli, Hanne F. Harbo, Tone Berge, Friedemann Paul, Alexander U. Brandt, Julio Saez-Rodriguez, Leonidas G. Alexopoulos, Elena H. Martinez-Lapiscina, Pablo Villoslada.

**Data curation:** Keith E. Kennedy, Nicole Kerlero de Rosbo, Maria Cellerino, Federico Ivaldi, Paola Contini, Raffaele De Palma, Hanne F. Harbo, Tone Berge, Steffan D. Bos, Einar A. Høgestøl, Synne Brune-Ingebretsen, Sigrid A. de Rodez Benavent, Alexander U. Brandt, Priscilla Bäcker-Koduah, Janina Behrens, Joseph Kuchling, Susanna Asseyer, Claudia Chien, Hanna Zimmermann, Seyedamirhosein Motamedi, Josef Kauer-Bonin, Melanie Rinas, Leonidas G. Alexopoulos, Magi Andorra, Sara Llufriu, Albert Saiz, Yolanda Blanco, Eloy Martinez-Heras, Elisabeth Solana, Irene Pulido-Valdeolivas.

**Formal analysis:** Keith E. Kennedy, Federico Ivaldi, Paola Contini, Tone Berge, Steffan D. Bos, Synne Brune-Ingebretsen, Sigrid A. de Rodez Benavent, Alexander U. Brandt, Julio Saez-Rodriguez, Leonidas G. Alexopoulos, Magi Andorra, Eloy Martinez-Heras, Irene Pulido-Valdeolivas, Jordi Garcia-Ojalvo.

**Funding acquisition:** Antonio Uccelli, Hanne F. Harbo, Tone Berge, Friedemann Paul, Julio Saez-Rodriguez, Leonidas G. Alexopoulos, Pablo Villoslada.

**Investigation:** Nicole Kerlero de Rosbo, Antonio Uccelli, Maria Cellerino, Federico Ivaldi, Paola Contini, Raffaele De Palma, Hanne F. Harbo, Tone Berge, Steffan D. Bos, Einar A. Høgestøl, Synne Brune-Ingebretsen, Sigrid A. de Rodez Benavent, Friedemann Paul, Alexander U. Brandt, Priscilla Bäcker-Koduah, Janina Behrens, Joseph Kuchling, Susanna Asseyer, Michael Scheel, Claudia Chien, Hanna Zimmermann, Seyedamirhosein Motamedi, Josef Kauer-Bonin, Melanie Rinas, Leonidas G. Alexopoulos, Sara Llufriu, Albert Saiz, Yolanda Blanco, Eloy Martinez-Heras, Elisabeth Solana, Irene Pulido-Valdeolivas, Elena H. Martinez-Lapiscina, Jordi Garcia-Ojalvo, Pablo Villoslada.

**Methodology:** Nicole Kerlero de Rosbo, Antonio Uccelli, Federico Ivaldi, Paola Contini, Raffaele De Palma, Tone Berge, Steffan D. Bos, Einar A. Høgestøl, Sigrid A. de Rodez Benavent, Friedemann Paul, Alexander U. Brandt, Janina Behrens, Joseph Kuchling, Susanna Asseyer, Michael Scheel, Claudia Chien, Hanna Zimmermann, Seyedamirhosein Motamedi, Josef Kauer-Bonin, Julio Saez-Rodriguez, Melanie Rinas, Leonidas G. Alexopoulos, Magi Andorra, Sara Llufriu, Eloy Martinez-Heras, Irene Pulido-Valdeolivas, Elena H. Martinez-Lapiscina, Jordi Garcia-Ojalvo, Pablo Villoslada.

**Project administration:** Antonio Uccelli, Hanne F. Harbo, Friedemann Paul, Alexander U. Brandt, Julio Saez-Rodriguez, Jordi Garcia-Ojalvo, Pablo Villoslada.

**Resources:** Yolanda Blanco, Irene Pulido-Valdeolivas, Elena H. Martinez-Lapiscina.

**Software:** Keith E. Kennedy, Magi Andorra, Eloy Martinez-Heras, Jordi Garcia-Ojalvo.

**Supervision:** Antonio Uccelli, Hanne F. Harbo, Alexander U. Brandt, Leonidas G. Alexopoulos, Jordi Garcia-Ojalvo, Pablo Villoslada.

**Validation:** Hanne F. Harbo, Friedemann Paul, Michael Scheel, Julio Saez-Rodriguez, Leonidas G. Alexopoulos, Elena H. Martinez-Lapiscina, Jordi Garcia-Ojalvo, Pablo Villoslada.

**Visualization:** Keith E. Kennedy, Pablo Villoslada.

**Writing – original draft:** Keith E. Kennedy, Antonio Uccelli, Hanne F. Harbo, Tone Berge, Friedemann Paul, Alexander U. Brandt, Julio Saez-Rodriguez, Leonidas G. Alexopoulos, Jordi Garcia-Ojalvo, Pablo Villoslada.

**Writing – review & editing:** Pablo Villoslada.

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
