## [Decision Letter · Decision Letter 0]

18 May 2023

Dear Dr. Villoslada,

Thank you very much for submitting your manuscript "Multiscale networks in multiple sclerosis" for consideration at PLOS Computational Biology.

As with all papers reviewed by the journal, your manuscript was reviewed by members of the editorial board and by several independent reviewers. In light of the reviews (below this email), we would like to invite the resubmission of a significantly-revised version that takes into account the reviewers' comments.

We ask you to read carefully the reviewers' comments and revise the manuscript accordingly, both in the scientific content and in the presentation and format. One reviewer raised concerns on lack of validation of the analysis algorithms, and another reviewer is concerned about the quality and accesibility of the figues, which the editors agree.

We cannot make any decision about publication until we have seen the revised manuscript and your response to the reviewers' comments. Your revised manuscript is also likely to be sent to reviewers for further evaluation.

Sincerely,

Zhaolei Zhang

Academic Editor

PLOS Computational Biology

Thomas Leitner

Section Editor

PLOS Computational Biology

Reviewer's Responses to Questions

**Comments to the Authors:**

Reviewer #1: This paper presents a network analysis using multiomics data to discover biological pathways associated with multiple sclerosis (MS), ranging from SNPs to clinical phenotypes. The authors constructed multilayer networks that interconnected feature networks of individual omics from a designated cohort, including SNPs, genes & proteins, cell types, endophenotypes from MRIs and OCTs, as well as clinical phenotypes. They prioritized cross-layer pathways using Boolean simulation to identify critical paths, such as linking specific proteins to immune cell types to retinal nerve fiber layer thickness to MS severity. Some of these pathways were validated using single-cell data.

In general, the study presents an interesting application of network biology to understand MS biology. However, several details, particularly in the methods section, are insufficiently described, making it difficult to evaluate the rigor of this study. The rationale for network selection and construction is also unclear. Additionally, the manuscript requires further editing to improve clarity. Here, I just highlight some major concerns.

1. Multiomics data. It remains unclear whether the multiomics data were obtained from the same individuals in the cohort. It appears that certain omics layers heavily relied on public databases, which may not be specific to MS.

2. Network construction. How to interconnect individual networks using mutual information again is unclear. It is unclear how the Boolean simulation would be sensitive to mutual information cutoffs. Moreover, it is unclear why Figure 3 switched back to correlation instead of mutual information, considering that proteins and genes likely exhibit strong nonlinear relationships. In sum, the authors did not provide sufficient details about their methodology.

3. Gene regulatory network. It remains unclear how specific the public gene regulatory networks used in the study are to MS, particularly linking SNPs to genes. The underlying gene regulatory mechanisms for constructing edges would be more useful than mutual information, such as SNPs interrupting transcription factor bindings and chromatin interactions to regulate downstream target genes.

4. Biology. While MS is known to be related to immunology, other cell types, such as neurons, also play important roles. Given the network biology approach, neglecting such crucial cell types would be a significant miss.

5. Writing. The manuscript would benefit from harmonizing the writing style across different sections to improve overall clarity.

Reviewer #2: Overall:

This work is timely and speaks to an important goal in understanding disease mechanism: how to integrate across multiple layers of data and identify correlations across layers to make a causal inference across these. The work has promising conceptual strengths, which when executed rigorously, could help advance the field of systems biology of disease.

However, in the current form, the work provides insufficient information to help the reader evaluate the validity of the methods, and significance of the results.

I recommend that this paper be re-evaluated after major revisions to help the community better evaluate the significance of this work.

Strengths:

They use a multi-modal dataset in MS that is large by contemporary scales, considering all the layers profiled.They have data from five layers profiled, including clinical, genomic and imaging layers.

The work is innovative in proposing the use of mutual information and Boolean gating to create a multi-layered network and infer cross-layer correlation. This is an important methodological idea that needs to be developed (see recommendations for revision).

They attempt to validate the cross-layer edges identified by their Boolean analysis by linear regression between phosphoproteomic values and phenotypes scores.

Major revisions:

1. My biggest concern with this work is that the algorithm used to infer pathways seems to be completely new, and no benchmarking is shown to convince the reader that:

a. the method has a reasonable sensitivity/specificity and will indeed highlight true associations while not identifying false associations

b. The method works equally well on the diverse data types shown - SNPs, imaging data and clinical data have different data characteristics. How do we know it is valid to apply this strategy to all these diverse datatypes and get something meaningful?

c. The method accurately groups patients with similar clinical profiles together. E.g., Do patients with the same type of MS co-cluster together?

This comment applies to the mutual information networks and the Boolean flow inference work.

This method is the foundation of all inferences in this paper. The authors need to show a proof-of-concept of some kind, e.g., apply this new algorithm to an existing multi-modal dataset (e.g., TCGA breast cancer data) and show that the method captures known pathways (SNP > gene > protein > pathway > phenotype, at the least).

Without some kind of reference point, it is hard to believe the validity of this method and I would not recommend the publication of this work.

2. Although the authors attempt to validate pathway links using regression, there is no clear presentation of what the sensitivity and specificity of the validation is. Part of the problem is the presentation. The pathways identified are simply listed in bullet points in the text. The validation is limited to one indirect link (phosphoproteomics to phenotype, but multiple phenotypic variables are compared - multiple testing correction?), and to correlation. In its current state, too much effort is required on the reader’s part to understand how many of the pathways and links identified by their Boolean logic were validated and how to evaluate the accuracy of the method and the results.

I recommend:

Create a table in which you list all the pathways identified (same ones as mentioned in the body of the text). Then: 1) add a column to indicate which of these were validated by the regression analysis and which were not validated (yes/no), and 2) add a column to indicate whether the path is known in the literature (cite).

Add a Venn diagram showing the overlap between “known links in literature” and “predicted links”

Discuss these factors in the Discussion section.

Without this reference point for the reader, the presentation of this result makes it hard to evaluate what the added benefit of the multi-layer networks is.

3. Figures have detailed colourful networks without any guidance as to the message of each panel is.

a) Unclear what the message of Figure 1b-g is. There is no visible substructure. There is no attempt in the text to link network topology to disease features or subtypes. What is the take-home message of these panels?

b) Why are the nodes coloured by degree, what is the significance of this? Why aren't they connected by similarity?

4. Please revise the abstract to tell us: What is the major conclusion of this paper? What new pathway has been identified in this work that was validated? How has this work improved the mechanistic understanding of MS? Currently the abstract seems to simply say that some results were found, without providing broader context to place the significance of the work.

Minor:

1. Too many acronyms, hard to read the work. Please expand acronyms right in Table 1 instead of putting in the footer. First time you use any acronym, please use full form.

2. In all the figures, the node labels are too small to read and it is not possible to draw any inference from these. Please remove or make bigger.

3. The validation is shown for three proteins, but the abstract lists many more. Why is this? It seems inconsistent. Why not just talk about what you validated?

4. In the introduction and abstract the authors mention that systems-level function is important for complex diseases. This is not just true about diseases, it is true about any phenotype for a multi-layered organism. Suggest changing this to reflect the fact.

5. Visualization: The authors use different colours to represent the same variable (confusingly, this is node degree) for different data types. The colour choices also have no relationship to the position of the layer in the system (e.g., lighter is closer to genetic, darker is closer to higher-order phenotypes). While divergent colours are pleasing to the eye they do not contribute to the informative visual exploration of the data.

Reviewer #3: In this manuscript from Kennedy et al., the authors analyze multiple layers of omics and clinical data from a large prospective cohort of MS patients and healthy controls. They use network analyses to identify salient networks within layers. They then perform interesting and novel analyses to integrate information across layers. This represents an impressive dataset and analytical approach. However, I found the manuscript at times hard to understand and there are some major concerns that need to be addressed.

- As someone who is not an expert in network analyses, I had a lot of difficulty following many of the methods and approaches used. It would be helpful to have a less technical description of the analytical approaches used.

- I found many of the figures to be hard to follow. There are many figures with a lot of networks and connections between networks, but the figure sizes are so small and it’s hard to know what is important in each figure. Also, I have no idea what figure 3C is trying to show. Rather than showing a lot of network graphs, it would be helpful to have a different approach for visualizing the results so readers can better understand the magnitude of effects. For example, take a SNP identified in the analyses, and show the changes in genomics, proteomics, clinical outcomes as a function of the SNP carrier status.

- What do the authors mean by “genomics”? Is this based on RNA-seq data? If so, what tissue was this done on?

- What were the ethnic and/or ancestry compositions of the cohorts? The authors do not seem to take into account ancestry as a covariate in their analyses

- There seems to be no replication of their findings. The networks identified should be replicated in a second dataset. Alternatively, the authors could use half the dataset to identify networks and the other half of the dataset to validate findings

- The authors do not seem to account for treatment as a covariate in their analyses. Furthermore, the DMT given to patients may result in some reverse causation effects (i.e., an individual may have been selected to be on a given DMT based on their clinical outcome), which would greatly confound results.

**Have the authors made all data and (if applicable) computational code underlying the findings in their manuscript fully available?**

Reviewer #1: None

Reviewer #2: **No: **Data has been made available. Networks and pathways (results) are made available in github repos. But I couldn't see where code was available, maybe I missed it.

Reviewer #3: Yes

PLOS authors have the option to publish the peer review history of their article (what does this mean?). If published, this will include your full peer review and any attached files.

Reviewer #1: No

Reviewer #2: No

Reviewer #3: **Yes: **Michael Guo
---

## [Decision Letter · Decision Letter 1]

12 Dec 2023

Dear Dr. Villoslada,

We are pleased to inform you that your manuscript 'Multiscale networks in multiple sclerosis' has been provisionally accepted for publication in PLOS Computational Biology.

Before your manuscript can be formally accepted you will need to complete some formatting changes, which you will receive in a follow up email. A member of our team will be in touch with a set of requests. Please try to improve the quality of the figures on the finalized version of the manuscript.

Best regards,

Zhaolei Zhang

Section Editor

PLOS Computational Biology

Zhaolei Zhang

Section Editor

PLOS Computational Biology

Reviewer's Responses to Questions

**Comments to the Authors:**

Reviewer #1: The authors addressed my comments including a justification of using healthy gene regulatory networks as reference.

Reviewer #3: The authors have done a nice job during the revision process. The overall text was much easier to understand, which helped me gain a much better appreciation for the novelty of their work. I still find that the figures are hard to understand and the network graphs are a bit overwhelming. I do appreciate that the authors have provided GitHub links to higher resolution interactive versions of some of these plots. I would recommend that Figure 8 be removed, as I am unsure what the utility of this Venn Diagram is. This data could be much more effectively displayed in a table.

**Have the authors made all data and (if applicable) computational code underlying the findings in their manuscript fully available?**

Reviewer #1: None

Reviewer #3: Yes

PLOS authors have the option to publish the peer review history of their article (what does this mean?). If published, this will include your full peer review and any attached files.

Reviewer #1: No

Reviewer #3: **Yes: **Michael Guo

---

## [Editor Report · Acceptance letter]

23 Jan 2024

PCOMPBIOL-D-23-00317R1 

Multiscale networks in multiple sclerosis

Dear Dr Villoslada,

I am pleased to inform you that your manuscript has been formally accepted for publication in PLOS Computational Biology. Your manuscript is now with our production department and you will be notified of the publication date in due course.

With kind regards,

Timea Kemeri-Szekernyes
